# Elemental and water-insoluble organic carbon in Svalbard snow: A synthesis of observations during 2007–2018.

Christian Zdanowicz[1], Jean-Charles Gallet[2], Mats P. Björkman[3], Catherine Larose[4], Thomas Schuler[5], Bartłomiej Luks[6], Krystyna Koziol[7], Andrea Spolaor[8], Elena Barbaro[8,9], Tõnu Martma[10], Ward van Pelt[1], Ulla Wideqvist[11] and Johan Ström[11].

[1]Department of Earth Sciences, Uppsala University, Uppsala, 752 36, Sweden
[2]Norwegian Polar Institute, Tromsø, NO-9296, Norway
[3]Department of Earth Sciences, University of Gothenburg, 405 30 Gothenburg, Sweden
[4]Environmental Microbial Genomics, École Centrale de Lyon, Université de Lyon, 69134 Ecully, France
[5]Department of Geosciences, University of Oslo, 0316 Oslo, Norway
[6]Institute of Geophysics, Polish Academy of Sciences, 01-452 Warszawa, Poland
[7]Department of Analytical Chemistry, Gdańsk University of Technology, 80-233 Gdańsk, Poland
[8]Institute of Polar Sciences, ISP-CNR, 30170 Venice Mestre, Italy
[9]Department of Environmental Sciences, Informatics and Statistics, Ca' Foscari University of Venice, 30172 Mestre, Italy
[10]Department of Geology, Tallinn University of Technology, 19086 Tallinn, Estonia
[11]Department of Environmental Science, Stockholm University, 106 91 Stockholm, Sweden

*Correspondence to*: Christian Zdanowicz (christian.zdanowicz@geo.uu.se)

**Abstract.**

Light-absorbing carbonaceous aerosols emitted by biomass or fossil fuel combustion can contribute to amplify Arctic climate warming by lowering the albedo of snow. The Svalbard archipelago, being near to Europe and Russia, is particularly affected by these pollutants, and improved knowledge of their distribution in snow is needed to assess their impact. Here we present and synthesize new data obtained on Svalbard between 2007 and 2018, comprising measurements of elemental (EC) and water-insoluble organic carbon (WIOC) in snow from 37 separate sites. We used these data, combined with meteorological data and snowpack modelling, to investigate the variability of EC and WIOC deposition in Svalbard snow across latitude, longitude, elevation and time. Overall, EC concentrations ($C_{snow}^{EC}$) ranged from <1.0 to 266.6 ng g$^{-1}$, while WIOC concentrations ($C_{snow}^{WIOC}$) ranged from <1 to 9426 ng g$^{-1}$, with the highest values observed near Ny-Ålesund. Calculated snowpack loadings ($L_{snow}^{EC}$, $L_{snow}^{WIOC}$) on glaciers surveyed in Spring 2016 were 0.1 to 2.6 mg m$^{-2}$ and 2 to 173 mg m$^{-2}$, respectively. The median $C_{snow}^{EC}$ and the $L_{snow}^{EC}$ on those glaciers were close to or lower than those found in earlier (2007–09), comparable surveys. Both $L_{snow}^{EC}$ and $L_{snow}^{WIOC}$ increased with elevation and snow accumulation, with dry deposition likely playing a minor role. Estimated area-averaged snowpack loads across Svalbard were 1.1 mg EC m$^{-2}$ and 38.3 mg WIOC m$^{-2}$ for the 2015-16 winter. An ~11-year long dataset of spring surface snow measurements from central Brøgger Peninsula was used to quantify the interannual variability of EC and WIOC deposition in snow. In most years, $C_{snow}^{EC}$ and $C_{snow}^{WIOC}$ at Ny-Ålesund (50 m a.s.l.) were 2–5 times higher than on the nearby Austre Brøggerbreen glacier (456 m a.s.l.), and the median EC/WIOC in Ny-Ålesund was 6 times higher, suggesting a possible influence of local EC emission from Ny-Ålesund. While no long-term trends between 2011 and 2018 were found, $C_{snow}^{EC}$ and $C_{snow}^{WIOC}$ showed synchronous variations at Ny-Ålesund and Austre Brøggerbreen. When compared with data from other circum-Artic sites obtained by comparable methods, the median $C_{snow}^{EC}$ on Svalbard falls between that found in central Greenland (lowest) and those in continental sectors of European Arctic (northern Scandinavia, Russia and Siberia; highest), which is consistent with large-scale patterns of BC in snow reported by surveys based on other methods.

## 1 Introduction

Light-absorbing black carbon (BC) aerosols that are transported to Arctic latitudes can lower the albedo of snow/ice-covered surfaces on which they are deposited, thereby enacting a positive feedback that amplifies climate warming (Bond et al., 2013).

The Svalbard archipelago, owing to its proximity to the European and Russian mainland, is particularly affected by BC emissions from fossil fuel combustion (FF; heating, gas flaring, etc.) and biomass burning (BB; e.g., agricultural or forest fires). Source attribution using carbon isotopes and atmospheric transport modelling indicates that BC associated with pollution haze events at the Zeppelin Observatory on Spitsbergen include both BB and FF contributions, the latter being proportionally more important in winter than summer (Winiger et al., 2015, 2019). Quantifying the impact of BC deposition on the Arctic surface albedo requires knowledge of its concentrations, spatial distribution and variability in snow and ice. These data may also serve to verify the efficacy of ongoing and future measures to curb emissions of short-lived climate forcing aerosols, such as BC, that impact the Arctic (AMAP, 2015; Stohl et al., 2015). On Svalbard, reconnaissance surveys of BC in snow were carried out in 1984–85 by Noone and Clark (1988) and in 2007 by Doherty et al. (2010). This was followed in 2007–09 by more detailed investigations of the distribution of BC across the archipelago (Forsström et al., 2009, 2013). Localized studies have also been carried out near Longyearbyen (Aamaas et al, 2011; Khan et al., 2017) and Ny-Ålesund (Sihna et al., 2018; Jacobi et al., 2019). In addition, two ice cores recovered from the Lomonosovfonna and Holtedahlfonna icefields (Spitsbergen) have provided insights into longer-term variations in BC deposition on Svalbard (Ruppel et al., 2014, 2017; Osmont et al., 2018).

Here we present and synthesize new observational data which document the variability of BC in snow across Svalbard in terms of latitude, longitude, altitude and time. These data were gathered through field investigations conducted between 2007–18 on both Spitsbergen and Nordaustlandet (**Fig. 1**). The datasets were developed from surveys carried out on glaciers of Svalbard in the springs 2016 and 2017, and from an 11-year long program of snow sampling for EC on central Brøgger Peninsula on northwestern Spitsbergen. The Spring 2016 survey included some of the sites previously visited in 2007–09 by Forsström et al. (2009, 2013), thus allowing for comparisons after a ~decadal interval. All data presented in this study were obtained by the thermo-optical transmittance method (TOT), which quantifies separately the more refractory and volatile carbon mass fractions present in water-insoluble, particulate material filtered from melted snow (Cavalli et al., 2010; see below). Following Petzold et al. (2013), we designate the more refractory mass fraction as elemental carbon (EC; see **Table 1** for a list of abbreviations used in this paper). The more volatile fraction represents carbon that evolved primarily from water-insoluble organic carbon (WIOC), but may also include some contributions from the oxidation of carbonate particles (carbonatic carbon; CC). While the WIOC data are not the main focus of our study, they are also presented, as such data are sparse for Arctic snow. We used our datasets in combination with meteorological snowpack modelling, to (i) describe the spatial distribution of EC and WIOC deposited on Svalbard glaciers, (ii) estimate their mass loading in the winter snowpack and how it relates to spatial variations in snow accumulation, and (iii) describe the interannual variability of EC and WIOC concentration in snow on Brøgger Peninsula between 2007 and 2018. Lastly, we place our findings in a broader pan-Arctic perspective by comparison with a compilation of published data obtained between 2002 and 2018 by comparable methods.

## 2 Material and methods

### 2.1 Field sampling

### 2.1.1. Spring 2016 glacier survey

Part of the dataset presented here was produced following a comprehensive, coordinated survey of the physical, chemical and microbiological properties of the Svalbard seasonal snowpack carried out at the end of the 2015–16 winter by individuals from multiple institutions (see acknowledgments). In total, 22 sites were sampled between 4 and 29 April 2016 on 7 individual glaciers of Spitsbergen and Nordaustlandet (**Fig. 1**). Snowpits were sampled at three different elevations in the upper, middle and lower reaches of each glacier, the snow depth increasing with altitude (e.g., Pramanik et al., 2019). In the text and figures, specific snowpit locations are identified by a letter and number code, for e.g., HB3 = Hansbreen glacier, snowpit #3 (**Table 2**, with additional details in **Table S1**). Glacier sites were targeted partly for logistical reasons (ease of access by snowmobile),

but also because sampling supraglacial snow avoided the heterogeneities in snow properties that may arise from interactions with vegetation and/or different substrates (e.g., wet vs. dry tundra soils). In addition, the selected glacier sites were at elevations of 102 to 1193 m a.s.l., which span ~65 % of the maximum relief in Svalbard (1713 m; **Fig. 2**).

In advance of the field campaign, standardized protocols were developed for the measurement of important snow physical properties (e.g., density, temperature) and the collection of samples for a variety of analyses, including EC, WIOC, and stable oxygen isotope ratios ($\delta^{18}$O) in water. These protocols are documented in Gallet *et al.* (2018), and details relevant for this paper summarized hereafter. Snow sampling was performed in snowpits excavated down to the hard, icy firn surface representing the previous year's late summer ablation surface (in the accumulation zone of glaciers), or to the underlying bare ice surface (in the ablation zone). All snowpits were located well away from point sources of contamination (e.g., field camps), were accessed by foot from at least 100 m, and personnel doing the sampling wore protective, non-particulating suits, gloves, and face masks, and employed pre-cleaned plastic or stainless steel tools. The snow accumulation of each snowpit ($h_{SWE}$, in water equivalent; w.e.) was calculated from discrete density measurements. After recording the physical properties of snow strata, large volume snow samples (~5 L each) were collected from the top 5 cm of the snowpack, and, where snowpack depth allowed for it, at 50-cm depth intervals beneath. The surface samples were collected to quantify EC concentrations in snow ($C_{snow}^{EC}$) at depths where light absorption by carbonaceous particles has the largest impact on snow albedo (Marks and King, 2013). The deeper samples were used to estimate the total column mass loading of EC and WIOC ($L_{snow}^{EC}$, $L_{snow}^{WIOC}$) in the seasonal snowpack. Quantification of $C_{snow}^{EC}$ (and, concurrently, of $C_{snow}^{WIOC}$) in layers from discrete snowfall events was not feasible, owing to the large snow volume required to achieve a sufficient particulate carbon mass for TOT analysis. All snow samples were double-bagged in sterile low-density polyethylene bags and returned frozen to a location where they were subsequently melted and filtered. Depending on logistics, this was done either at the Polish Polar Station Hornsund operated by the Polish Academy of Sciences, or at the Norwegian Polar Institute (NPI) facilities in Ny-Ålesund (Sverdrup station), Longyearbyen (UNIS) or Tromsø. A total of 89 samples were obtained from all 22 sites.

Analysis of downscaled climatological fields from the European Centre for Medium-Range Weather Forecasts (ERA-Interim; Dee et al., 2011) over Svalbard show that surface temperatures in the 2015–16 winter exceeded the 30-year climatological normals for the 1981–2010 period by 2 to 6 °C, with the largest anomalies observed in the northeastern part of the archipelago (**Fig. S1a**). Total winter precipitation also exceeded 1981–2010 normals by 0.2 to 0.7 m w.e. over much of central and northern Spitsbergen (**Fig. S1b**). These unusual conditions arose partly owing to an extreme winter warming and precipitation event associated with a southerly air intrusion over Spitsbergen that occurred in late December 2015 (Binder et al., 2017; Kim et al., 2017; Maturilli et al., 2017). The implications of these climatological circumstances for the interpretation of our snow survey data are discussed later.

**2.1.1. Spring 2017 glacier survey**

We also report additional $C_{snow}^{EC}$ and $C_{snow}^{WIOC}$ measurements in surface snow collected at 17 sites on 5 glaciers in northwestern Spitsbergen (*n* = 26; see **Table 1,** with additional details in **Table S2**). These samples were collected by NPI staff during other field research activities, but were handled and analyzed in the same manner as those of the Spring 2016 glacier survey.

**2.1.2. Surface snow monitoring (2007-2018), Brøgger Peninsula**

Surface snow layers have been sampled for EC measurements by NPI staff at several sites on Brøgger Peninsula between 2007 and 2018 (**Fig. 1**, see enlargement in **Fig. S2** for details). The first of these sites is in the accumulation zone of the glacier Austre Brøggerbreen (ABG; 78.87° N, 11.92° E, 456 m a.s.l.), which was accessed by snowmobile from Ny-Ålesund. The other sites are in the outskirts of Ny-Ålesund, one ~80 m southeast of NPI's Sverdrup station (78.92° N, 11.93° E, ~50 m a.s.l.), the other near the Gruvebadet Atmospheric Laboratory (78.92° N, 11.89° E, ~50 m a.s.l.). Sampling was carried out at approximately weekly intervals by the NPI permanent staff at Sverdrup station, whenever their work schedule made it possible,

and when safe snowmobile driving conditions (e.g., proper visibility, firm surface) allowed access to Austre Brøggerbreen. Because of these restrictions, the snow samples could not always be collected immediately after snow fall events. Over the ~11-year period considered, a total of 201 samples were collected between February and June, 86 % of which were taken in the spring months (March–May), April being the most represented month ($n = 44$). Methods for field sample collection were the same as those described above for the April 2016 survey. Sample collection was limited to the top 5 cm of the snowpack (occasionally deeper). These data provide long-term estimates of the interannual variability of $C_{snow}^{EC}$ and $C_{snow}^{WIOC}$ in Svalbard against which results of the April 2016 survey (and others) can be compared.

## 2.2 EC and WIOC analyses

### 2.2.1. Laboratory procedures

The snow samples collected routinely by NPI staff near Ny-Ålesund and on Austre Brøggerbreen were processed following the protocol described in Forsström *et al.* (2009). Briefly, snow was transferred into pre-cleaned, 1 L borosilicate glass beakers and melted in a microwave oven. At 800 W power, the typical melting time was ~10 min kg⁻¹. The meltwater was then filtered through pre-ashed, 47-mm diameter quartz microfiber filters (**Fig. S3**). For the spring 2016 and 2017 glacier surveys, the large number of samples obtained, combined with the need to process samples at various field stations (to avoid risky transport), made the use of microwave ovens impractical. Instead, these samples were melted at room in temperature inside their closed bags. Melting the 5-L samples took ~24-36 hours, depending on the exact volume and density of the material. The meltwater was drained periodically and filtered as it was being produced, i.e. filtration was done in steps, as the samples melted, so the total duration of melting does not equate the time any water was left standing in the bags. In the end, the bags were rinsed with Milli-Q water and the rinse water was also filtered, and the added water volume accounted for.

All the filters produced from the snow samples were air-dried at room temperature overnight, stored in sterile petri dishes, and later sent to the Department of Environmental Science of Stockholm University. There, EC/WIOC analysis was performed using a Sunset Laboratory carbon aerosol analyzer (Sunset Laboratory Inc., Forest Grove, USA), following the European Supersites for Atmospheric Aerosol Research thermal evolution protocol (EUSAAR_2; Cavalli *et al.*, 2010). One benefit of this protocol is that it minimize biases in the EC-WIOC split that may be caused by charring and pyrolysis of organic compounds on filters (e.g., Cheng et al., 2014). On each day of measurements, filters prepared with standard sucrose solutions were used to calibrate the instrument for WIOC measurements (e.g., Pantedeliadis *et al.*, 2015). There is presently no corresponding, widely-accepted standard for calibration of EC by thermo-optical analysis. However Svensson *et al.* (2018) showed, using the same instrument and with the same thermal protocol as used in our study, that predictable and reproducible results can be obtained for filters prepared from diluted aqueous suspensions of NIST-2975 diesel soot, or from chimney soot.

For the measurements on our samples, a 1 x 1 cm² square section was used from each filter to determine separately the particulate EC and WIOC mass density, or loading, on each filter ($L_{filt}^{EC}$, $L_{filt}^{WIOC}$), from which their mass concentrations in snow ($C_{snow}^{EC}$, $C_{snow}^{WIOC}$) were calculated based on the volume of meltwater filtered, following:

$$C_{snow}^{EC} = \frac{L_{filt}^{EC} A_{\text{filt}}}{V_{meltwater}}$$
(1)

$$C_{snow}^{WIOC} = \frac{L_{filt}^{WIOC} A_{\text{filt}}}{V_{meltwater}}$$
(2)

where $A_{\text{filt}}$ is the total area of the filter on which particulate matter was collected, and $V_{\text{meltwater}}$ the volume of meltwater filtered.

Calculated values of $C_{snow}^{EC}$ and $C_{snow}^{WIOC}$ are reported in ng EC g$^{-1}$ snow, and ng WIOC g$^{-1}$ snow, respectively. Blank filters ($n =$ 6) had particulate carbon loadings below the limit of detection (LOD) of the carbon analyzer, so no blank correction was applied to the data.

     During the first three years of the NPI snow monitoring program in Ny-Ålesund area and on the glacier Austre Brøggerbreen, the filters had been analyzed on the carbon aerosol analyzer using the NIOSH-5040 thermal protocol (Birch

2003). Subsequent inter-comparison studies established that this protocol gave values of $L_{filt}^{EC}$ that were lower than than those obtained by the EUSAAR-2 protocol by an average factor of two (Cavalli et al., 2010). Therefore, and as discussed earlier in Forsström et al. (2013), these data were later corrected by this factor to homogenize the results with those obtained by the EUSAAR-2 protocol. Since WIOC is hundreds of time more abundant than EC in snow, the correction has a negligible impact on those results, which were left uncorrected.


**2.2.2. Methodological uncertainties**

     Several factors can contribute to uncertainties in TOT analyses of filters produced from melted snow. While some of these were previously investigated and quantified for EC (as discussed below) they remain largely unquantified for WIOC. In the absence of such information, we have used the same estimates of uncertainties pertaining to EC for WIOC, but our

confidence in the latter is lower. Some uncertainties in EC quantification arise irrespective of particle composition. These include heterogeneous distribution of EC in the snowpack, loss of particles to container walls, undercatch during filtration, and uneven loading of particles on the filters. The natural variability of $C_{snow}^{EC}$ in snow at the meter scale due to snowpack heterogeneity ($\sigma_{sh}$) was previously estimated by Forsström et al. (2009) and by Svensson et al. (2013) in Arctic snow, and we therefore used their results here. The loss of EC particles to glass containers walls was experimentally found by Svensson et

al. (2009) to be negligible for storage times < 48 hours (in a sample containing containing ~35 ng g$^{-1}$ EC). Here, we have assumed this to also be the case for the samples melted in bags, although we acknowledge that losses may occur, which would have a larger impact for samples with very low EC content. We used results from Amaas et al. (2011), Svensson et al. (2013) and Forsström et al. (2013) to estimate the possible effect of filtration undercatch, since these studies used the same methods as our own, and were based on double-filtration of real snow samples, rather than on "standards"(e.g., diesel soot) which may

have properties different than those of actual snow particulates (Torres et al., 2014). The aforementioned studies gave estimates of undercatch that ranged between 18-35 %, with a median of 22 %. Undercatch mostly affects the smallest particles, which contribute little to the total BC particle mass (Lim et al., 2014). Furthermore, the formation of particle aggregates (BC alone, or BC-dust) as snow ages likely modify their initial size distribution (Schwarz et al., 2012; Svensson et al., 2018). The snow samples analyzed in our study were variously aged, and included relatively fresh, near surface snow, and subsurface snow

layers which were affected by partial melting-refreezing during winter thaw events (a frequent occurrence on Svalbard). Since we could not quantify how these differences would affect filtration efficiency, we opted not to apply any systematic undercatch corrections to our EC (and WIOC) data.

     The uncertainty in EC quantification arising from uneven filter loading ($\sigma_f$) was evaluated from paired measurements on separate 1-cm$^2$ filter sections prepared from the Spring 2016 snow survey samples ($n = 87$). The coefficients

of variation (CV) between paired measurements of both $L_{filt}^{EC}$ and $L_{filt}^{WIOC}$ were found to scale up with the particle mass density on the filters. Based on these results, the median value for $L_{filt}^{EC}$ was 19 % (8 % for $L_{filt}^{WIOC}$), with an interquartile range of 7 to 35 % (4 to 19 % for $L_{filt}^{WIOC}$). These estimates also account for the carbon analyzer's precision, which contributes to observed spread of results. By combining the various sources of uncertainty listed above, we estimated a median coefficient of variation (CV) of ~40 % for both $C_{snow}^{EC}$ and $C_{snow}^{WIOC}$.

## 2.2.3. Possible effects of dust in $C_{snow}^{EC}$ and $C_{snow}^{WIOC}$

A total of 31 snow filters obtained from 7 glaciers surveyed in Spring 2016 (35 % of samples) were found to have faint to pronounced yellow-pink or grey-brown coloration, likely indicating the presence of k-feldspars and/or oxides which are commonly found in cryoconites, although carbonates may also be present on these filters (see notes in Supplemental dataset). The presence of mineral dust particles can lead to biases in the separation of EC and WIOC by the thermo-optical method, owing to several competing effects: (a) $CO_2$ released from CC and incorrectly detected as EC, (b) pyrolysis of organic carbon by oxide minerals, which may lead to EC underestimation, and (c) the formation of EC-dust aggregates, which is more likely to occur in aged sub-surface snow than in relatively fresh surface snow (Wang et al., 2012; Lim et al., 2014; Kuchiki et al., 2015).

There are, to our knowledge, no reliable data on the mineral dust content in Svalbard snow, but various published data on trace element analyses suggest typical values between a few tens of ppb (ng $g^{-1}$) and a few ppm (µg $kg^{-1}$) based on typical crustal proportions (Casey, 2012; Singh et al., 2015; Thomas et al., 2020). Carbonate minerals are only a minor dust constituent in the wintertime aerosol over Svalbard, and thus unlikely to contribute much to deposition in snow (Weinbruch et al., 2012; Moroni et al, 2015). However carbonate rocks outcrop in many areas of Svalbard (e.g., central Spitsbergen, Brøgger Peninsula), are common constituents in some cryosols (4 to 37 % mass; Szymański et al., 2015; Hanaka et al., 2019) and are found in cryoconites on at least some Spitsbergen glaciers (Hodson et al., 2010; Langford et al., 2011). Wind-blown dust deflated from local soils or sediments is therefore a potential source of carbonates and other mineral particles in Svalbard snow.

The EUSARR-2 protocol used in this study minimizes the effect of effect of carbonate minerals by causing CC to evolve into $CO_2$ during the He-mode of TOT analysis, thus being detected as WIOC, rather than EC (Cavalli et al., 2010). Since WIOC is hundreds of times more abundant than EC, the effect on the measured WIOC concentrations is comparatively very small. Some authors have suggested pre-treatments to remove carbonates on filters (e.g., Evangeliou et al., 2018), but as discussed in Svensson et al. (2018), these treatments may actually raise more issues than they solve, and so they were not applied in our study. Furthermore, some snow samples that were collected early during the NPI snow monitoring program near Ny-Ålesund were tested at Stockholm University for the effect that carbonate removal by acid fumigation had on EC quantification by TOT, but the resulting changes in thermograms were judged too minor to justify applying this procedure routinely.

With regards to the effect of oxides, Lim et al. (2014) reported probable artefacts due to such minerals in TOT analyses of alpine snow samples containing 1-10 µg $kg^{-1}$ of dust, but did not give the magnitude of the associated errors on the measured EC and WIOC concentrations. Nor did they find any systematic correlation between dust amounts and the presence or absence of such artefacts in TOT thermograms. Hence at present, there are simply no firm grounds on which to base any error estimates or corrections. This will require dedicated research, which is outside the scope of our paper.

Concerning the effect of EC-dust aggregates, Wang et al. (2012) showed that in snow with >20 ng $g^{-1}$ EC, the formation of aggregates can lead to underestimation of EC by the TOT method, unless samples are sonicated prior to filtration (sonication, however, may increase filtration undercatch). The largest underestimation found in that study, for a single sample, was 20 % (80 ng $g^{-1}$ prior to sonication, 100 ng $g^{-1}$ after). For EC < 20 ng $g^{-1}$ (all other samples), the effect of sonication (hence, presumably, of aggregates) was negligible. In our own data, EC > 20 ng $g^{-1}$ occur in < 3 % of glacier samples, but in ~30 % of snow samples from near Ny-Ålesund (Sverdrup and Gruvebadet sites). The glacier snow samples that produced colored filters were typically found in layers near the base of snowpits, suggesting windblown dust dispersion and deposition in the autumn when the ground is only partially snow-covered. In 11 of the colored filters, the measured $L_{filt}^{EC}$ were noticeably lower than in filters from snow layers immediately above, and in 6 filters the $L_{filt}^{EC}$ was < LOD. Thus may be due to the effect of dust on TOT measurements, but it is presently impossible to confirm.

With regards to samples collected at Sverdrup or Gruvebadet near Ny-Ålesund, we hypothesized above that the more frequent, relatively elevated EC levels in surface snow near Ny-Ålesund compared to those on the glacier Austre Brøggerbreen (400 m higher) were due to local combustion emissions. If locally-emitted dust (e.g., mobilized by road traffic in or near the town) caused under-estimation of EC in local snow due to dust-BC aggregates being formed, then the difference between the median EC in snow near Ny-Ålesund and that in snow at Austre Brøggerbreen may in fact be larger than we surmised, which would in fact reinforce our hypothesis.

### 2.2.4. Calculation of EC and WIOC loadings

For the snowpits excavated on glaciers in Spring 2016, we computed mass loadings of EC ($L_{snow}^{EC}$) and WIOC ($L_{snow}^{WIOC}$) in the seasonal snowpack following:

$$L_{snow}^{EC} = \sum_{i=1}^{2} (C_{snow}^{EC})_i \rho_i z_i \tag{3}$$

$$L_{snow}^{WIOC} = \sum_{i=1}^{n} (C_{snow}^{WIOC})_i \rho_i z_i \tag{4}$$

where $L_{snow}^{EC}$ and $L_{snow}^{WIOC}$ are in mg m$^{-2}$, $\rho_i$ is the mean density of snow layer $i$ in kg m$^{-3}$, $z_i$ its thickness in m, and $n$ the number of discrete layers. For samples which yielded $C_{snow}^{EC}$ <1 ng g$^{-1}$ we assigned a value of 0.5 ng g$^{-1}$ (half the LOD) in order to compute snowpack loadings (see below). An estimated error on individual density measurements ($\sigma_\rho$) of $\pm$ 6 % was used (Conger and McClung, 2009; Proksch et al., 2016), and the meter-scale variability of snow layer density at spatial scales of 1 to 100 m$^2$ was assumed to be on the order of 5 %, after Koenig et al. (2016). Combining uncertainties on $C_{snow}^{EC}$ and $C_{snow}^{WIOC}$ with these errors yields a median CV of ~30 % for $L_{snow}^{EC}$ and $L_{snow}^{WIOC}$ ($n$ = 22 snowpits).

### 2.3 Determination of δ¹⁸O in snow meltwater

The stable isotope ratio of oxygen ($^{18}O$:$^{16}O$) in melted snowpit samples collected in April 2016 was used to detect evidence of warming events associated with large autumn or winter snowfalls, that could help to interpret the $L_{snow}^{EC}$ and $L_{snow}^{OC}$ data. The measurements were made at the Institute of Geology of Tallinn's University of Technology, Estonia, using a Picarro model L2120-i water isotope analyzer (Picarro Inc., Sunnyvale, USA) (Lis et al., 2008). Results are reported in the standard delta notation δ¹⁸O relative to the Vienna Standard Mean Ocean Water. The analytical precision was ±0.1 ‰.

### 2.4 Supporting data

### 2.4.1 Surface meteorological observations

Automated weather stations (AWS) were operated on 6 glaciers sampled during the April 2016 survey (**Table S3**). These stations were situated close to the estimated equilibrium line altitude (ELA) of the glaciers, and provided hourly recordings of air temperature and ultrasonic soundings of snow surface height changes that were used to interpret snowpit stratigraphic data, in particular the timing of snow accumulation and of snowmelt events. Data from the AWSs were supplemented with records from Longyearbyen and the airport in Ny-Ålesund obtained from the Norwegian Meteorological Institute, and from the Polish Polar Station Hornsund (N 77.00°, E 15.11°, 9 m a.s.l.).

### 2.4.2 Snowpack modeling

Owing to the scarcity of direct precipitation measurements across Svalbard, reconstructing the snowpack accumulation history is challenging, and estimates from snowpits, probing and radar can only fill some of the spatial and temporal gaps. This

difficulty can be partly circumvented by using the output of a snowpack model forced with meteorological observations (e.g., Jacobi et al., 2019). In this study, we use a coupled energy balance-snow model (van Pelt et al., 2012), which has recently been
employed to investigate glacier and snow conditions across Svalbard (van Pelt et al., 2019). The model includes subroutines for the surface energy balance as well as internal snowpack processes (e.g., densification, melt-freeze events) that makes it possible to simulate the evolution of the seasonal snowpack (thickness and internal structure) for individual 1x1 km grid cells over Svalbard. The snow model routine simulates subsurface density, temperature and water content, while accounting for vertical water transport, liquid water storage, refreezing and runoff (Marchenko et al. 2017; Van Pelt et al. 2019). Fresh snow
density is described by a temperature- and wind-dependent function (Van Kampenhout et al. 2017), while snow densification is the sum of destructive metamorphism, compaction by overburden pressure and compaction by drifting snow (Vionnet et al. 2012). Snow scouring and redistribution by wind is not accounted for, however. Layered snow properties are modelled with a vertical resolution of 1 cm. The performance of the model was estimated by Van Pelt et al. (2012) by comparing the predicted winter surface mass balance $b_w$ with the water-equivalent snow accumulation ($h_{SWE}$) measured over a network of reference
stakes. Across all glaciers, the RMSE was between 0.12 and 0.33 m w.e., and averaged 0.23 m w.e., with a mean bias of zero.

In this paper, we used the model to (1) simulate the snowpack evolution at some of the glacier sites sampled during the Spring 2016 survey, and (2) quantify the variability of snowfall and snow cover over Brøgger Peninsula for the period 2008–18 during which surface snow was sampled for EC and WIOC. For the Spring 2016 survey snowpits, simulations were limited to sites located close to or above the local equilibrium line altitude (identified in **Table S1**). As in van Pelt et al. (2019),
the model was forced with downscaled, 3-hourly meteorological fields generated with the High Resolution Limited Area Model (HIRLAM, version 6.4.2; Reistad et al., 2009). Since our objective (1) was to estimate the relative timing of snow accumulation, and not absolute snow depth, precipitation at all the modelled glacier sites was locally calibrated (scaled with a factor) to assure matching between modelled and observed snow depths at the time of observation (April 2016). For objective (2), however, no such adjustments were made.

**3 Results**

Descriptive statistics of $C_{snow}^{EC}$ and $C_{snow}^{WIOC}$ for all samples analyzed in this study are summarized in **Table 2**. The probability distributions of $C_{snow}^{EC}$ and $C_{snow}^{WIOC}$ are positively-skewed (right-tailed), therefore we use medians ($\tilde{C}_{snow}^{EC}$, $\tilde{C}_{snow}^{WIOC}$) as measures of their central tendency, but also report arithmetic and geometric means for comparisons with other published data. As both $C_{snow}^{EC}$ and $C_{snow}^{WIOC}$ are left-censored by the analytical LOD, the median and mean were estimated by replacing values < 1 ng g$^{-1}$
with 0.5 x LOD (0.5 ng g$^{-1}$), following Hornung and Reed (1990), while the geometric mean was estimated by the beta factor method of Ganser and Hewett (2010). Values of $C_{snow}^{EC}$ and $C_{snow}^{WIOC}$ < LOD are, however, included in plots (see below) to provide an as complete as possible description of our data. Overall, $C_{snow}^{EC}$ ranged from <1.0 to 266.6 ng g$^{-1}$, while $C_{snow}^{WIOC}$ ranged from <1 to 9426 ng g$^{-1}$. The highest $C_{snow}^{EC}$ (>50 ng g$^{-1}$) were measured in spring surface snow near Ny-Ålesund (Sverdrup and Gruvebadet sites). The $\tilde{C}_{snow}^{EC}$ at these two sites over the period 2007–18 (9.8 ng g$^{-1}$) was ~1.4 to 4.0 times higher
than in surface snow layers at glacier sites (2.4–6.8 ng g$^{-1}$). At most sampling sites, EC accounted for <30 % (most typically, <5 %) of the total mass of particulate carbon (EC+WIOC) in snow, except near Ny-Ålesund, where it accounted for up to 61 %.

**3.1. Snow on glaciers, Spring 2016 and 2017 surveys**

The spatial variations of $C_{snow}^{EC}$ and $C_{snow}^{WIOC}$ across the glacier sites surveyed in Spring 2016 are summarized on **Fig. 3** and **4**.
Data from samples collected on glaciers of northwestern Spitsbergen in Spring 2017 are shown on these plots for comparison, but are limited to surface snow layers. In the Spring 2016 snowpack ($n$ = 22 sites), $C_{snow}^{EC}$ ranged from <1.0 to 22.7 ng g$^{-1}$, with $\tilde{C}_{snow}^{EC}$ = 2.9 ng g$^{-1}$, while $C_{snow}^{WIOC}$ ranged from 12 to 550 ng g$^{-1}$, with $\tilde{C}_{snow}^{WIOC}$ = 49 ng g$^{-1}$. At nearly all sites, the $C_{snow}^{EC}$ in surface

layers (top 5–10 cm) was larger than the weighted mean for the whole snowpack (**Fig. S4**). This was not the case for $C_{snow}^{WIOC}$, which showed no systematic enrichment in surface layers relative to the bulk of the snowpack. In the Spring 2016 snowpack, $C_{snow}^{EC}$ >10 ng g$^{-1}$ were only found on glaciers of southern Spitsbergen (Hornsund, HB and Werenskioldbreen, WSB; max. 22.8 ng g$^{-1}$). The snowpack on these two glaciers also had the most snow layers with $C_{snow}^{WIOC}$ >100 ng g$^{-1}$. Comparing the snow surface layers sampled on glaciers in northwestern Spitsbergen in the springs of 2016 (April) and 2017 (April and May) suggests higher springtime EC and WIOC deposition in 2017. On the glacier Kongsvegen (KVG), which was sampled in both years, the $\tilde{C}_{snow}^{EC}$ and $\tilde{C}_{snow}^{WIOC}$ were 4 and 12 times higher in 2017, respectively.

No statistically meaningful differences in $\tilde{C}_{snow}^{EC}$ were found in the Spring 2016 snowpack between the northwestern, central and southern sectors of Spitsbergen (**Fig. 3**, range: 1.1–2.4 ng g$^{-1}$; Kruskal-Wallis test, $p$ >0.1). However the snowpack on Austfonna (AF) on the island of Nordaustlandet, northeastern Svalbard, had a significantly lower $\tilde{C}_{snow}^{EC}$ (1.1 ng g$^{-1}$) than on glaciers of northwestern and southern Spitsbergen. This held true even if values of $C_{snow}^{EC}$ <1 ng g$^{-1}$ were excluded. The $C_{snow}^{WIOC}$ in the Spring 2016 snowpack were generally lowest on Lomonosovfonna (LF; central Spitsbergen) and on Austfonna (AF), and highest on southern Spitsbergen glaciers (HB and WSB). Binning the 2016 snowpit data by elevation (**Fig. 4**; 200 m bins) showed no meaningful differences in $\tilde{C}_{snow}^{EC}$ (2.2–2.8 ng g$^{-1}$) or $\tilde{C}_{snow}^{WIOC}$ (50.3–103.0 ng g$^{-1}$) over the ~1100 m altitude range of the 22 glacier sampling sites (Kruskal-Wallis test, $p$ >0.1). The calculated $L_{snow}^{EC}$ in the spring 2016 snowpack were between 0.1 and 2.6 mg m$^{-2}$ with a median of 0.7 mg m$^{-2}$ (mean 0.9 mg m$^{-2}$), while $L_{snow}^{WIOC}$ were between 1.7 and 173.2 mg m$^{-2}$, with a median of 20.5 mg m$^{-2}$ (mean 41.2 mg m$^{-2}$). There were no discernible patterns of variation of $L_{snow}^{EC}$ or $L_{snow}^{WIOC}$ with respect to geographic location, but on most glaciers $L_{snow}^{EC}$ and/or $L_{snow}^{WIOC}$ increased with elevation along with $h_{SWE}$ (see section 4.2.2.).

### 3.2. Surface snow monitoring (2007-2018), Brøgger Peninsula

Variations of $C_{snow}^{EC}$ and $C_{snow}^{WIOC}$ measured in the surface snow of central Brøgger Peninsula between 2007 and 2018 are shown on **Fig. 5**. In most months, $C_{snow}^{EC}$ was between 1 and 100 ng g$^{-1}$, and $C_{snow}^{WIOC}$ between 10 and 1000 ng g$^{-1}$. For years in which snow samples from both the glacier Austre Brøggerbreen (ABG) and Ny-Ålesund were obtained, the range of variations on ABG ($C_{snow}^{EC}$: <1–45.1 ng g$^{-1}$; $C_{snow}^{WIOC}$: <1–1076.1 ng g$^{-1}$) overlapped with that near Ny-Ålesund ($C_{snow}^{EC}$: <1–266.5 ng g$^{-1}$; $C_{snow}^{WIOC}$: <1–7250.3 ng g$^{-1}$; 2 outliers excluded; **Fig. 5a,b**). However in most years, $\tilde{C}_{snow}^{EC}$ near Ny-Ålesund was 2–5 times greater than at ABG. Likewise, $\tilde{C}_{snow}^{WIOC}$ was frequently 2–3 times larger. There were significant interannual variations in springtime $\tilde{C}_{snow}^{EC}$ (range: 0.4–8.2 ng g$^{-1}$) and $\tilde{C}_{snow}^{WIOC}$ (range: 1.8–691.4 ng g$^{-1}$) between 2007 and 2018. The highest $\tilde{C}_{snow}^{EC}$ and $\tilde{C}_{snow}^{WIOC}$ occured in the Spring of 2017, and the lowest in the Spring of 2014. Depending on site, $\tilde{C}_{snow}^{EC}$ in 2017 was 23–27 times higher than in 2014, and $\tilde{C}_{snow}^{WIOC}$ was 146–217 times higher, the largest differences being observed in the snow near Ny-Ålesund. Furthermore, in years in which samples were obtained from both ABG and Ny-Ålesund, the variations were clearly synchronous and coherent at these two sites, which are separated by ~5.5 km and ~400 m in elevation (**Fig. 5c**). The ratio of $\tilde{C}_{snow}^{EC}$ between Ny-Ålesund and ABG tended to increase with $\tilde{C}_{snow}^{EC}$ at both sites, as did the ratio of $\tilde{C}_{snow}^{WIOC}$ between these sites with $\tilde{C}_{snow}^{WIOC}$ (**Fig. 5d**). The EC/WIOC in snow varied between 0.01 and 0.42, and, in years when the comparison was possible, the variations in springtime EC/WIOC on ABG (2007–18 median: 0.08) tracked those at Ny-Ålesund (2010–18 median: 0.10) (**Fig. 5e**).

## 4 Discussion

### 4.2. Snow on glaciers, Spring 2016 and 2017 surveys

### 4.2.1. Spatial patterns in $C_{snow}^{EC}$ or $C_{snow}^{WIOC}$

The Spring 2016 survey showed no discernible zonal or latitudinal gradient of $C_{snow}^{EC}$ or $C_{snow}^{WIOC}$ across Svalbard. As noted earlier, only on Austfonna (AF) was $\tilde{C}_{snow}^{EC}$ significantly lower than in some sectors of Spitsbergen. This contrasts with findings from surveys made in the springs of 2007–09, in which $\tilde{C}_{snow}^{EC}$ on AF snow was either comparable to, or larger than, that in

central or northwestern Spitsbergen (Forsström et al., 2009, 2013). Based on data from sites where direct comparisons with the 2007–09 surveys are possible, $\tilde{C}_{snow}^{EC}$ in the seasonal snowpack varies, from year to year, by at least one order of magnitude, and sometimes more (**Fig. 3**). This was also evident for $\tilde{C}_{snow}^{EC}$, but also for $\tilde{C}_{snow}^{WIOC}$, in the surface snow layers sampled on
KVG, northwestern Spitsbergen, in Spring 2016 and 2017.

Our ability to detect possible spatial patterns in either $\tilde{C}_{snow}^{EC}$ or $\tilde{C}_{snow}^{WIOC}$ across Svalbard is presently limited by (1) the large variability in the observed data in both space and time (spread of individual observations), (2) differences in sampling density across geographic sectors of the archipelago, with central Spitsbergen being under-sampled due to its remoteness, and (3) uncertainties in the TOT method. Limitations due to (1) and (2) above can only be overcome through repeated surveys with
a more even even spatial coverage. We estimated the possible contribution of the methodological uncertainties on estimates of $\tilde{C}_{snow}^{EC}$ through a Monte Carlo approach in which surrogate data were generated from the $C_{snow}^{EC}$ measured in the Spring 2016 survey, assuming that the latter have normal distributions of errors with a CV of ± 40 %, as discussed above (section 2.2.2.). Results (**Fig. S5-S6**) showed that in samples (groups of observations) for which the spread of individual measurements of $C_{snow}^{EC}$ is relatively small, for example that from northeastern Svalbard (Austfonna; AF), the uncertainty on $\tilde{C}_{snow}^{EC}$ arising from
method-related errors is nearly as large as that which results from the overall spread of individual observations (shown as the notches about medians on the boxplots in **Fig. 3**). In other samples, such as that from central Spitsbergen (Lomonosovfonna; LF), the spread of observations far exceeds the expected uncertainty due to method-related errors. The same result was obtained for $\tilde{C}_{snow}^{WIOC}$ although, as stressed earlier, methodological uncertainties on these data are more poorly constrained than those for $\tilde{C}_{snow}^{EC}$. Hence, while reducing methodological uncertainties in the TOT analyses is certainly desirable, it may not be sufficient
to confidently detect spatial variations in $\tilde{C}_{snow}^{EC}$ (or $\tilde{C}_{snow}^{WIOC}$) across Svalbard, given the inherent large variability in the snowpack.

### 4.2.2. Spatial patterns in $L_{snow}^{EC}$ or $L_{snow}^{WIOC}$

Comparing results of the Spring 2016 glacier survey with the 2007–09 survey results from Forsström et al. (2013) showed that the $L_{snow}^{EC}$ in the late winter snowpack across Svalbard can vary by at least two orders of magnitude between years. For example, $L_{snow}^{EC}$ at the summit of Holtedahlfonna in northwestern Spitsbergen (site HDF3; elev. 1119 m a.s.l.) was 1.1 mg m$^{-2}$ in April 2016, which is 70% lower than the 3.7 mg m$^{-2}$ calculated in April 2008 at the same location (Forsström et al., 2013). For their part, Ruppel et al. (2017) estimated an annual mean $L_{snow}^{EC}$ of 10 mg m$^{-2}$ using snow samples and a firn core from
Holtedahlfonna spanning ~8 years (2006–14). The corresponding mean $L_{snow}^{EC}$ in the late winter (end April) snowpack could be less than half of this value (~5 mg m$^{-2}$), but the high interannual variability in net snow accumulation at this site (Pramanik et al., 2014; Van Pelt and Kohler, 2015), and the uncertainty in the chronology of the firn core makes such an estimate tentative at best. Differences between our estimates of $L_{snow}^{EC}$ and those from the 2007–09 surveys probably reflect, to a large extent, the variability of atmospheric EC transport and deposition between years and seasons, but also in space (local scale; Svensson et
al., 2013).

### 4.2.3. Variations of $L_{snow}^{EC}$ and $L_{snow}^{WIOC}$ on glaciers with elevation and snow accumulation

The estimated $L_{snow}^{EC}$ and $L_{snow}^{WIOC}$ in the Spring 2016 snowpack were generally largest at higher elevations on glaciers, where
snow accumulation is greater. For the winter 2015–16 snowpack, we modelled the relationship between $L_{snow}^{EC}$ and $h_{SWE}$ across all snowpits by robust least-squares linear regression (**Fig. 6a**; R$^2$ = 0.91; RMSE = 0.3). A linear model applied to $L_{snow}^{WIOC}$ against $h_{SWE}$ gave a poorer fit (**Fig. 6b**; R$^2$ = 0.51; RMSE = 8), owing to the much greater scatter in the $L_{snow}^{WIOC}$ data. Total least-squares ("type 2") regression models were also tested to account for the error on $h_{SWE}$ values arising from the snowpack model

uncertainties, and the results were closely comparable. Our calculated $L_{snow}^{EC}$ for sites with low $h_{SWE}$, such as those on the lower reaches of glaciers exposed to wintertime katabatic winds, likely underestimate both dry- and wet-deposited EC owing to wind scouring of the snowpack. Nevertheless, the intercept of our linear model for $L_{snow}^{EC}$ suggests that the contribution of dry deposition to the winter 2015-16 EC mass accumulation on Svalbard glaciers was likely negligible. In this respect, our findings are consistent with those of Sinha et al. (2018) based on direct observations of BC deposition at Ny-Ålesund in the winter 2012–13, but they contrast with those of Jacobi et al. (2019), who postulated a ~50 % contribution of BC dry deposition on the glacier Kongsvegen (our site KVG3, elev. 672 m a.s.l.) in Spring 2012, using calculations of deposition fluxes constrained by measurements in melted snowpack.

We applied the linear models for $L_{snow}^{EC}$ and $L_{snow}^{WIOC}$ from **Fig. 6** to a map of late winter (30 April) $h_{SWE}$ generated with the snowpack model in order to project the geographic pattern of EC and WIOC accumulation across the whole of Svalbard for the winter 2015–16 (**Fig. 7**). The $h_{SWE}$ data used for this purpose were extracted from the output presented in Van Pelt et al. (2019). Summing the predicted values for $L_{snow}^{EC}$ and $L_{snow}^{WIOC}$ across the land grid provides estimates of the total aerosol mass that accumulated in the snowpack. The area-averaged loads were 1.1 ± 0.1 mg EC m$^{-2}$ and 38.3 ± 0.2 mg WIOC m$^{-2}$. These figures translate to monthly mean accumulation rates of ~0.1 mg EC m$^{-2}$ mo$^{-1}$, and ~4.8 mg WIOC m$^{-2}$ mo$^{-1}$, respectively, over the period of snow accumulation from 1 Sept. 2015 and 30 April 2016.

**4.2.4. Relative timing of EC and WIOC accumulation on glaciers, winter 2015-16.**

Using the snowpack model, we also estimated the relative contributions to $L_{snow}^{EC}$ and $L_{snow}^{WIOC}$ made by each of the snowpack layers sampled in the accumulation zone of the 7 glaciers surveyed in Spring 2016. The number of layers sampled varied from 4 on Austre Lovénbreen (ALB3) and Lomonosovfonna (LF3), to 7 on Werenskioldbreen (WSB3). On-site surface height soundings by AWSs at several glaciers (**Fig. S7**) indicate that snow accumulation in the 2015–16 winter was more or less equally divided between the autumn period leading to the late December 2015 snowstorm, and the months that followed up to mid-/late April 2016, when the snowpits were sampled. The snowpack model simulations, forced with downscaled HIRLAM precipitation data, gave similar results (**Fig. 8**). The AWSs also show that the December 2015 storm saw winter temperatures on nearly all glaciers rise above 0°C for several days, the warming being largest in southern Spitsbergen. Clear evidence for this was found in a >0.2 m thick icy snow layer at mid-depth in the snowpack on Hansbreen (site HB3). The depth of the layer is in good agreement with that predicted by the snowpack model at this site, showing that the simulation provides a reasonable estimate of local surface conditions. Icy layers also occurred in the lower half of the seasonal snowpack on other glaciers, but none of these could be unambiguously ascribed to the late December 2015 storm period.

The relative timing of EC and WIOC accumulation, inferred from the snowpack model chronology, varied considerably between glaciers (**Fig. 8**). On Austfonna (AF), ~80 % of the EC and WIOC was found in snow layers estimated to have been deposited in or after December 2015. On glaciers of northwestern and central Spitsbergen (ALB, KVG, HDF and LF), the accumulation sequence was more variable and differed between EC and WIOC. On the glaciers Hansbreen (HB) and Werenskioldbreen (WSB) in southern Spitsbergen, most of the EC and WIOC was contained in the deep layers of the seasonal snowpack, estimated to have been deposited prior to January 2016. Surface meteorological records from the Polish Polar Station Hornsund and from an AWS on WSB show that several large snowfall events occurred in this area during the autumn of 2015, as well as some thaw events (**Fig. S7**). The stratigraphy of snowpits excavated on HB and WSB in April 2016 also showed clear evidence of melt-freeze events in the early part of the 2015–16 winter (e.g., site HB3; **Fig. 8**). Also visible in these snowpits were multiple positive excursions in δ$^{18}$O (i.e. shifts to less negative values) indicative of snowfall events presumably associated with relatively moist and warm southerly air intrusions over Spitsbergen. Conceivably, the relatively elevated $C_{snow}^{EC}$ and $C_{snow}^{WIOC}$ in the deeper snowpack layers on HB and WSB might have resulted from a few large autumn or early winter snowfall events, as previously observed for wet deposition of SO$_4^{2-}$, NO$_3^-$ and WIOC on glaciers near Hornsund

(Kühnel et al., 2013; Kozioł et al., 2019). Alternatively, or concurrently, meltwater percolation during surface thaws (or rain-on-snow) may have redistributed or concentrated some of the more hydrophilic EC and WIOC into snow layers near the base of the snowpack (Aamaas et al., 2011; Xu et al., 2012).

### 4.3. Surface snow monitoring (2007-2018), Brøgger Peninsula

Two features of the ~11-year record of $C_{snow}^{EC}$ and $C_{snow}^{WIOC}$ from Ny-Ålesund and from Austre Brøggerbreen (ABG) (**Fig. 5**) are noteworthy, in that they reveal some spatio-temporal patterns in the data. One such feature is the mean difference in $\tilde{C}_{snow}^{EC}$ and $\tilde{C}_{snow}^{WIOC}$ between the two sites (as noted in section 3.2), the other is the synchronous interannual variations in $C_{snow}^{EC}$ and $C_{snow}^{WIOC}$ at both sites. The generally higher $\tilde{C}_{snow}^{EC}$ and $\tilde{C}_{snow}^{WIOC}$ near Ny-Ålesund compared to ABG suggests that a gradient in atmospheric EC and WIOC deposition exists between these sites in late winter and spring, which may be a function of relative distance from the coast and/or elevation, Ny-Ålesund and ABG being separated by ~5.5 horizontal km and an altitude difference of ~400 m. One plausible explanation for such a gradient is that owing to its greater remoteness, the accumulation area of ABG is less influenced by local aerosol emissions from Ny Ålesund (EC and WIOC) or from coastal waters (WIOC). This was in fact the rationale for the choice of Zeppelin Mountain (474 m. a.s.l.) as the site for a permanent atmospheric observatory (Braathen et al., 1990). The diesel-fueled power plant in Ny Ålesund is a candidate source of EC and semi-volatile organic compounds (precursors for WIOC), as is wintertime vehicular traffic of cars, snowmobiles, and aircrafts (Shears et al., 1998; Robinson et al., 2007; Dekhtyareva et al., 2016). In a previous survey in April 2008, Aamaas et al. (2011) could find no detectable local EC pollution in coastal snow within 20 km of Ny-Ålesund, although none was sampled downwind of the settlement. Our data, however, suggest that that winter/spring surface snow near Ny-Ålesund is commonly enriched in EC when compared to snow deposited higher up on ABG. Ice-free open water areas or frost flowers on sea ice are other potential sources of particulate WIOC (e.g., microbes, diatoms, plankton, exopolymers from biofilms) during autumn and winter, some of which are likely deposited in snow by settling or through ice nucleation (Bowman and Deming, 2010; Campbell et al., 2018; Karl et al., 2019). Stable inversion layers established by strong surface radiative cooling could trap aerosols emitted from Ny-Ålesund itself or from nearby open waters or sea ice, leading to enhanced concentrations of these aerosols in coastal surface snow during these periods. The frequent occurence of near-surface temperature inversions below ~500 m in winter and spring months were previously shown to enhance the concentration of airborne sulfate aerosols (from fuel combustion and/or marine sources) in Ny Ålesund relative to the Zeppelin Observatory (Dekhtyareva et al., 2018). It thus seems plausible that a similar effect may apply to EC and WIOC aerosols. If a gradient in EC and WIOC deposition to snow does exist between Ny Ålesund and ABC, it would not necessarily persist in all winter or spring months. For example, Aamaas et al. (2011) measured a $\tilde{C}_{snow}^{EC}$ of 6.6 ng g$^{-1}$ near Ny-Ålesund in March 2008, which was very close to the mean of 6.3 ng g$^{-1}$ on ABG snow during the same month.

Coherent interannual variations in $\tilde{C}_{snow}^{EC}$ and $\tilde{C}_{snow}^{WIOC}$ between Ny Ålesund and the accumulation area of ABG (**Fig. 5c**) were described in section 3.2. The amplitude of these variations was frequently larger than the uncertainty of medians for individual seasons (estimated using the same Monte Carlo approach described in section 4.2.1). It thus seems unlikely that such a coherent pattern arose by chance alone, for e.g., owing to random methodological errors. Several possible explanations could account for it. One is that year-to-year variations in springtime long-range transport and deposition of EC and WIOC aerosols in this part of Brøgger Peninsula impact the surface snow chemistry in Ny Ålesund and at ABG in a similar way. Another is that in some years, local meteorological conditions promote more efficient dispersion and/or scavenging of aerosols emitted from Ny Ålesund (EC and WIOC) and nearby coastal waters (WIOC), such that deposition in surface snow layers is enhanced at both Ny Ålesund and ABG simultaneously. These two explanations are not mutually exclusive. However the mean springtime ratio of $\tilde{C}_{snow}^{EC}$ between Ny Ålesund and ABG was observed to increase in years with relatively higher $\tilde{C}_{snow}^{EC}$, as did the ratio of $\tilde{C}_{snow}^{WIOC}$ between the two sites in years with higher $\tilde{C}_{snow}^{WIOC}$ (**Fig. 5d**). This argues against the temporal variations being due to interannual changes in the mean thickness of the springtime atmospheric boundary layer, since if this was the

case, one would expect the differences (ratios) of $\tilde{C}_{snow}^{EC}$ and $\tilde{C}_{snow}^{WIOC}$ to decrease, not inrease, in years when more aerosols from low-levels sources reach up to ABG, as was clearly not the case.

The median EC/WIOC in snow at Ny-Ålesund and ABG was <0.10 prior to 2010, but rose to 0.43 and 0.20, respectively, in 2015, and declined after 2016 (**Fig. 5e**). The seasons with highest median EC/WIOC (2013–15) were also those with lowest $\tilde{C}_{snow}^{EC}$ and $\tilde{C}_{snow}^{WIOC}$, which suggests that meteorological and/or other conditions prevailed in these seasons which limited atmospheric deposition of WIOC and EC in snow, WIOC being more affected than EC. Possible causal factors include sea ice cover or sea-surface winds, which partly modulate emissions of marine organic aerosol (e.g., Kirpes et al., 2019), or katabatic winds from Kongsfjorden, that can affect the thermal stratification of boundary layer air in winter months (Esau and Repina, 2012; Maturilli and Kayser, 2017).

Between 2008 and 2018 (the years in which snow sampling was most thorough), $\tilde{C}_{snow}^{EC}$ on Brøgger Peninsula varied by up to 35 ng g$^{-1}$, and $\tilde{C}_{snow}^{WIOC}$ by up to 689 ng g$^{-1}$. However, there were no significant trends in either $\tilde{C}_{snow}^{EC}$ or $\tilde{C}_{snow}^{WIOC}$ over the whole period (Mann-Kendall test; $p \gg 0.05$). Data from the Spring 2016 glacier survey (**Fig. 6**), as well as previous studies (Bourgeois and Bey, 2011; Browse et al., 2012) suggest that wet deposition is the predominant mode of EC deposition in Arctic snow. To investigate the possible role of snowfall on EC deposition on Brøgger Peninsula, we compared $\tilde{C}_{snow}^{EC}$ in surface layers for March, April and May (MAM) 2008–18 with simulated monthly snowfall anomalies over this region over the same period. However, no significant correlation was found.

## 4.4 Comparison of $C_{snow}^{EC}$ in pan-Arctic perspective

**Figure 10** shows the range of $C_{snow}^{EC}$ in winter/spring Svalbard snow measured in this study, compared with data from other circum-Arctic or subarctic sites sampled between 2005 and 2018 (see figure caption for data sources). All data in this comparison were obtained by thermo-optical measurements on snow filters using either the NIOSH-5040 or the EUSAAR-2 protocols (**Table S4**). The $C_{snow}^{EC}$ data obtained by the NIOSH-5040 were multipled by a factor of 2, as discussed earlier (section 2.2.1.) Laboratory-specific variations in the analytical protocols between studies could still account for some of the inter-site differences shown in **Fig. 10**. An indication of the possible spread of results that could arise from such variations is shown on the figure, based on method intercomparison studies by Panteliadis et al. (2015) and Bautista et al. (2015) for EC in aerosols. The pattern of EC distribution in winter/spring snow across the circum-Arctic shows the lowest $\tilde{C}_{snow}^{EC}$ in central Greenland (<1 ng g$^{-1}$), and the highest in northern mainland Scandinavia (~15–30 ng g$^{-1}$ depending on site) and in eastern Russia and central Siberia (35–66 ng g$^{-1}$). The $\tilde{C}_{snow}^{EC}$ in Svalbard snow fall within these extremes.

Previous, pan-Arctic scale surveys of BC in snow were conducted between 2007–09 (Doherty et al., 2010) and between 2012–16 (Mori et al., 2019). In the 2007-09 survey, the mass concentrations of BC in snow (denoted $C_{BC}^{est}$) was estimated optically by absorption spectrophotometry performed on filters (Integrating Sphere/Integrating Sandwich method; Grenfell et al., 2010), while in the 2012–16 surveys, the mass concentration of refractory BC (denoted $C_{MBC}$) was measured by laser-induced incandescence using a Single Particle Soot Photometer (SP2; Stephens et al., 2003). An error analysis by Doherty et al. (2010) indicated that individual measurements of $C_{BC}^{est}$ were subject to total uncertainties < 50 %, while Mori et al. (2019) estimated a mean uncertainty of ± 20 % on individual values of $C_{MBC}$, based on results of reproducibility tests. As in the present study, the uncertainty of reported medians values for groups of samples in any given region are expected to be considerably lesser than that of individual measurements. Irrespective of the methods used, results from the 2007–09 and 2012–16 surveys both showed that the highest BC oncentrations in Arctic snow were found in northwestern Russia and Siberia, followed by northern mainland Scandinavia, and the lowest occured in central Greenland, while concentrations in Svalbard fell between those in these regions (see Fig. 1 in Mori et al., 2019, and Fig. 2 in Dou and Xiao, 2016). Thus, the spatial pattern of $\tilde{C}_{snow}^{EC}$ shown in our data compilation (**Fig. 10**) is in broad agreement with findings from earlier surveys made within the past two decades.

## 5 Summary and conclusions

We have presented a large dataset of observations of atmospheric EC and WIOC deposited in snow on the archipelago of Svalbard, made between 2007 and 2018. These data will contribute to augment the existing body of observational data presently available for the validation of long-range transport and deposition models of BC in Arctic snow (Torseth et al., 2019). The spatial snow survey conducted across 22 glacier sites in Spring 2016 was one of the most extensive and detailed carried out on Svalbard, and allows direct comparisons with the surveys by Forsström et al. (2009, 2013), made nearly 10 years earlier. Across all glacier sites, $C_{snow}^{EC}$ in the snowpack ranged from <1.0 to 22.7 ng g$^{-1}$ (median 1.9 ng g$^{-1}$), while $C_{snow}^{WIOC}$ ranged from 12 to 550 ng g$^{-1}$ (median 49 ng g$^{-1}$). The calculated $L_{snow}^{EC}$ were between 0.1 and 2.6 mg m$^{-2}$ (median 0.7 mg m$^{-2}$), while $L_{snow}^{OC}$ were between 2 and 173 mg m$^{-2}$ (median 20 mg m$^{-2}$). The $\tilde{C}_{snow}^{EC}$ and $L_{snow}^{EC}$ in 2016 were comparable or lower than those found in spring 2007–09 glacier snow, but no clear spatial gradients could be identified across the archipelago. Both $L_{snow}^{EC}$ and $L_{snow}^{WIOC}$ were found to increase with elevation and $h_{SWE}$. Using these relationships, we estimated the area-averaged accumulation of EC and WIOC over the whole of Svalbard to be ~0.1 mg EC m$^{-2}$ and ~3.8 mg WIOC m$^{-2}$ for the winter 2015–16 (September to April). The relationship between $L_{snow}^{EC}$ and $h_{SWE}$ also point to dry EC deposition in snow being minor compared to wet deposition. The accumulation of EC and WIOC in the snowpack was inferred to be equally distributed over the winter 2015–16 at most glacier sites.

The set of EC and WIOC measurements made in surface snow on Brøgger Peninsula in 2007–18 is one of the longest such datasets available from the Arctic. During this period, the range of $C_{snow}^{EC}$ and $C_{snow}^{WIOC}$ near Ny-Ålesund (50 m a.s.l.) overlapped with that at Austre Brøggerbreen (456 m a.s.l.). However, $\tilde{C}_{snow}^{EC}$ and $\tilde{C}_{snow}^{WIOC}$ near Ny-Ålesund were, in most years, ~2–5 times higher than on Austre Brøggerbreen, which suggests the existence of seasonal gradient in EC and WIOC deposition between these sites. While no long-term trends were detected over the period 2007–18, $\tilde{C}_{snow}^{EC}$ and $\tilde{C}_{snow}^{WIOC}$ showed synchronous interannual variations between the snow sampling sites, the largest ones occurring near Ny-Ålesund. Further investigations of winter/spring micro- to mesoscale meteorological conditions are needed to clarify what the apparent gradient and synchronous variations in $\tilde{C}_{snow}^{EC}$ and $\tilde{C}_{snow}^{WIOC}$ between Ny-Ålesund and Austre Brøggerbreen might imply about the dynamics of atmospheric EC and WIOC deposition in snow at these sites. Simultaneously measurements of eBC in air and falling snow at both sites might also provide answers. Extending the surface snow monitoring program for EC and WIOC on Austre Brøggerbreen would allow to test the robustness of these findings presented here. Finally, we note that the methodological uncertainties on the determination of $C_{snow}^{WIOC}$ remain poorly constrained compared to those for of $C_{snow}^{EC}$, and this needs to be addressed in future dedicated studies.

*Data availability.*

The data presented in this article can be downloaded from https://doi.pangaea.de/10.1594/PANGAEA.918134.

*Author contributions.*

JCG, MB, CL, BL, TS, CZ and others initiated the April 2016 survey. JCG oversaw the snow sampling program on Brøgger Peninsula. UL and JC performed the EC and WIOC analyses, and TM the δ$^{18}$O analyses. WvP carried out the snow model simulations. CZ wrote the manuscript, with contributions from all co-authors.

*Competing interests.*

The authors declare that they have no conflict of interest.

*Acknowledgements.*

The April 2016 survey on Svalbard was funded through a grant from the Svalbard Science Forum (RIS 10472) to J.-C. Gallet and others, and surface snow monitoring on Brøgger Peninsula was conducted by the Norwegian Polar Institute (NPI) with support from Svalbardmiljøfonds and from the Swedish Research Council Formas project "Black and White" (grant 2006-00210 to J. Ström and others). Logistical support for field work in 2016 was provided by NPI, the University of Oslo, the Swedish Research Council and the Swedish Polar Secretariat, the Ice and Climate and the Environment research group at Uppsala University, the Institute of Geophysics of the Polish Academy of Sciences (PAS), the Centre for Polar Studies at the University of Silesia (USi), the Polish Ministry of Science and Higher Education (statutory activities 3841/E-41/S/2020), and the Institut Paul-Emile Victor (France). A. Uszczyk (USi), D. Ignatiuk (Svalbard Integrated Arctic Earth Observing System), M. Grabiec and D. Kępski (PAS) participated in the snow sampling work near Horsund. Planning and collaborative work between the study co-authors was facilitated by funds from the Swedish Strategic Research Area initiative "Biodiversity and Ecosystem services in a Changing Climate" through Gothenburg University, the Gothenburg Air and Climate Network and the International Arctic Science Committee.

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

| Acronym or symbol | Units | Definition |
| --- | --- | --- |
| BC | | Black carbon: Light-absorbing, refractory particulate carbon aerosols emitted by the incomplete combustion of organic fuels (biomass or fossil fuels). |
| TOT | | Thermo-optical transmittance method used to analyze particulate carbon in snow |
| EC | | Elemental carbon: Refractory carbon fraction determined by TOT |
| WIOC | | Water-insolube organic carbon: Volatile carbon fraction determined by TOT |
| CC | | Carbonatic carbon released by oxidation of carbonate minerals |
| $C_{snow}^{EC}$ | ng g$^{-1}$ | Mass concentration of EC in snow determined by the TOT method |
| $C_{snow}^{WIOC}$ | ng g$^{-1}$ | Mass concentration of OC in snow determined by the TOT method |
| $\tilde{C}_{snow}^{EC}$ | ng g$^{-1}$ | Median value of $C_{snow}^{EC}$ |
| $\tilde{C}_{snow}^{WIOC}$ | ng g$^{-1}$ | Median value of $C_{snow}^{OC}$ |
| $L_{filt}^{EC}$ | μg cm$^{-2}$ | Mass loading of EC on filters determined by the TOT method |
| $L_{filt}^{WIOC}$ | μg cm$^{-2}$ | Mass loading of OC on filters determined by the TOT method |
| $h_{SWE}$ | cm | Snow depth expressed in water equivalent |
| $L_{snow}^{EC}$ | mg m$^{-2}$ | Mass loading of EC in the seasonal snowpack, based on measurements of $C_{snow}^{EC}$ |
| $L_{snow}^{WIOC}$ | mg m$^{-2}$ | Mass loading of OC in the seasonal snowpack, based on measurements of $C_{snow}^{OC}$ |
| PSAP | | Particle Soot Absorption Photometer |

**Table 1.** Main symbols and acronyms used in this paper. The various terms for black carbon (BC, EC, eBC) are as defined in Petzold et al. (2013).

## Northwestern Spitsbergen

| Sampling site | Lat. N | Lon. E | Elev. (m a.s.l.) | S | P |
|---|---|---|---|---|---|
| **Northwestern Spitsbergen** | | | | | |
| Kongsvegen (KVG) | | | | | |
| KVG3 | 13° 20' | 78° 45' | 672 | X | X |
| KVG2 | 13° 09' | 78° 47' | 534 | X | X |
| | 12° 58' | 78° 48' | 395 | X | |
| KVG1.5 | 12° 52' | 78° 49' | 326 | X | X |
| KVG1 | 12° 46' | 78° 50' | 226 | X | X |
| | 12° 29' | 78° 52' | 3 | X | |
| Holtedahlfonna (HDF) | | | | | |
| HDF3 | 12° 24' | 79° 08' | 1119 | X | X |
| HDF2 | 12° 32' | 79° 02' | 718 | X | X |
| | 78° 59' | 13° 28' | 642 | X | |
| HDF1 | 13° 18' | 78° 56' | 570 | X | X |
| Austre Brøggerbreen (ABG) | | | | | |
| | 78° 52' | 11° 55' | 456 | X | |
| Vestre Brøggerbreen (VBG) | | | | | |
| | 78° 54' | 11° 40' | 450 | X | |
| | 78° 54' | 11° 40' | 355 | X | |
| | 78° 54' | 11° 41' | 300 | X | |
| | 78° 55' | 11° 44' | 139 | X | |
| Austre Lovénbreen (ALB) | | | | | |
| ALB3 | 12° 11' | 78° 52' | 513 | X | X |
| ALB2 | 12° 10' | 78° 53' | 340 | X | X |
| ALB1 | 12° 08' | 78° 53' | 195 | X | X |
| Midtre Lovénbreen (MLB) | | | | | |
| | 11° 59' | 78° 52' | 403 | X | |
| | 12° 02' | 78° 53' | 297 | X | |
| | 12° 04' | 78° 54' | 87 | X | |

| Sampling site | Lat. N | Lon. E | Elev. (m a.s.l.) | S | P |
|---|---|---|---|---|---|
| Edithbreen (EBR) | | | | | |
| | 11° 11' | 78° 51' | 625 | X | |
| | 11° 45' | 78° 51' | 425 | X | |
| Ny Ålesund area | | | | | |
| Sverdrup | 78° 55' | 11° 56' | ~50 | X | |
| Gruvebadet | 78° 55' | 11° 54' | ~50 | X | |
| **Central Spitsbergen** | | | | | |
| Lomonosovfonna (LF) | | | | | |
| LF3 | 17° 26' | 78° 49' | 1193 | X | X |
| LF2 | 17° 09' | 78° 41' | 523 | X | X |
| LF1 | 17° 05' | 78° 38' | 223 | X | X |
| **Southern Spitsbergen** | | | | | |
| Werenskioldbreen (WSB) | | | | | |
| WSB3 | 15° 29' | 77° 06' | 528 | X | |
| WSB2 | 15° 26' | 77° 04' | 413 | X | |
| WSB1 | 15° 19' | 77° 05' | 166 | X | |
| Hansbreen (HB) | | | | | |
| HB3 | 15° 29' | 77° 07' | 396 | X | X |
| HB2 | 15° 38' | 77° 05' | 275 | X | X |
| HB1 | 15° 38' | 77° 03' | 102 | X | X |
| **Nordaustlandet** | | | | | |
| Austfonna (AF) | | | | | |
| AF3 | 24° 00' | 79° 50' | 785 | X | X |
| AF2 | 22° 50' | 79° 46' | 507 | X | X |
| AF1 | 22° 25' | 79° 44' | 336 | X | X |

**Table 2.** Snow sampling sites in Svalbard mentioned in this paper. Glacier site codes that include a number (e.g., KVG3) are those where samples were collected during the Spring 2016 survey. The "x" symbols indicate whether samples were collected from surface layers (S) and/or in snowpits (P). A few samples from Austre Brøggrbreen (ABG) were collected below 456 m. a.s.l. Further details on sampling sites are provided in **Table S1** and **S2**.

| | Spring 2016 glacier survey (22 sites) | | Spring 2017 glacier survey (17 sites) | Austre Brøggerbreen 2007–18 (1 site) | Sverdrup & Gruvebadet 2010–18 (2 sites) |
|---|---|---|---|---|---|
| | All layers | Surface | Surface | Surface | Surface |
| $C_{snow}^{EC}$ (ng g$^{-1}$) | | | | | |
| $n$ | 87 | 22 | 26 | 115 | 84 |
| $n^*$ | 68 | 21 | 25 | 87 | 73 |
| Minimum | <1.0 | <1.0 | 1.6 | <1.0 | <1.0 |
| Maximum | 22.7 | 22.7 | 25.4 | 45.2 | 266.6 |
| Median | 1.9 | 2.4 | 6.8 | 3.4 | 9.8 |
| Mean | 2.9 | 4.7 | 8.1 | 6.4 | 35.1 |
| Geo. mean | 1.9 | 3.2 | 5.9 | 2.9 | 8.8 |
| $C_{snow}^{WIOC}$ (ng g$^{-1}$) | | | | | |
| $n$ | 87 | 22 | 26 | 109 | 79 |
| $n^*$ | 87 | 22 | 26 | 108 | 78 |
| Minimum | 12 | 21 | 18 | <1 | <1 |
| Maximum | 550 | 550 | 3426 | 1076 | 9426 |
| Median | 49 | 49 | 355 | 54 | 92 |
| Mean | 88 | 80 | 491 | 165 | 299 |
| Geo. mean | 61 | 57 | 267 | 48 | 67 |
| % EC | | | | | |
| Minimum | <1 | 1 | <1 | <1 | <1 |
| Maximum | 21 | 21 | 10 | 36 | 61 |
| Median | 3 | 4 | 2 | 6 | 12 |
| Mean | 4 | 7 | 3 | 8 | 15 |
| Geo. mean | 2 | 5 | 2 | 5 | 8 |

**Table 3.** Descriptive statistics for $C_{snow}^{EC}$, $C_{snow}^{WIOC}$ and % EC in samples analyzed in this study. $n^*$ is the number of values > 1 ng g$^{-1}$. Two outliers with $C_{snow}^{EC}$ >1700 ng g$^{-1}$ were excluded from calculations.

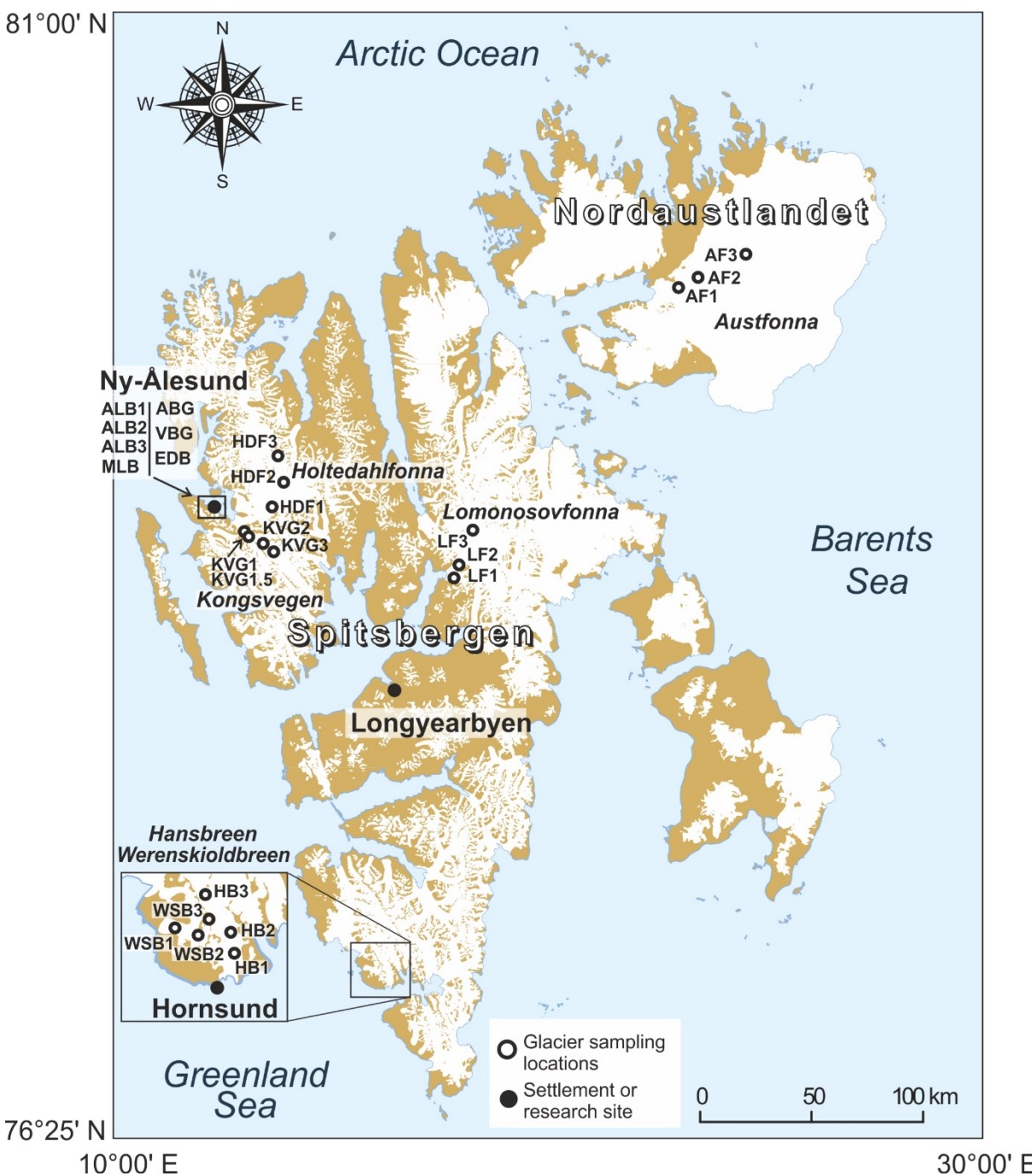

**Fig. 1.** Map of the Svalbard archipelago, showing the location of the snow sampling sites mentioned in this study (**Table 2**). Sampling sites on glaciers near Ny-Ålesund (framed area; sites ALB, MLB, ABG, VBG and EDB) are shown in an enlargement in **Fig. S2**.

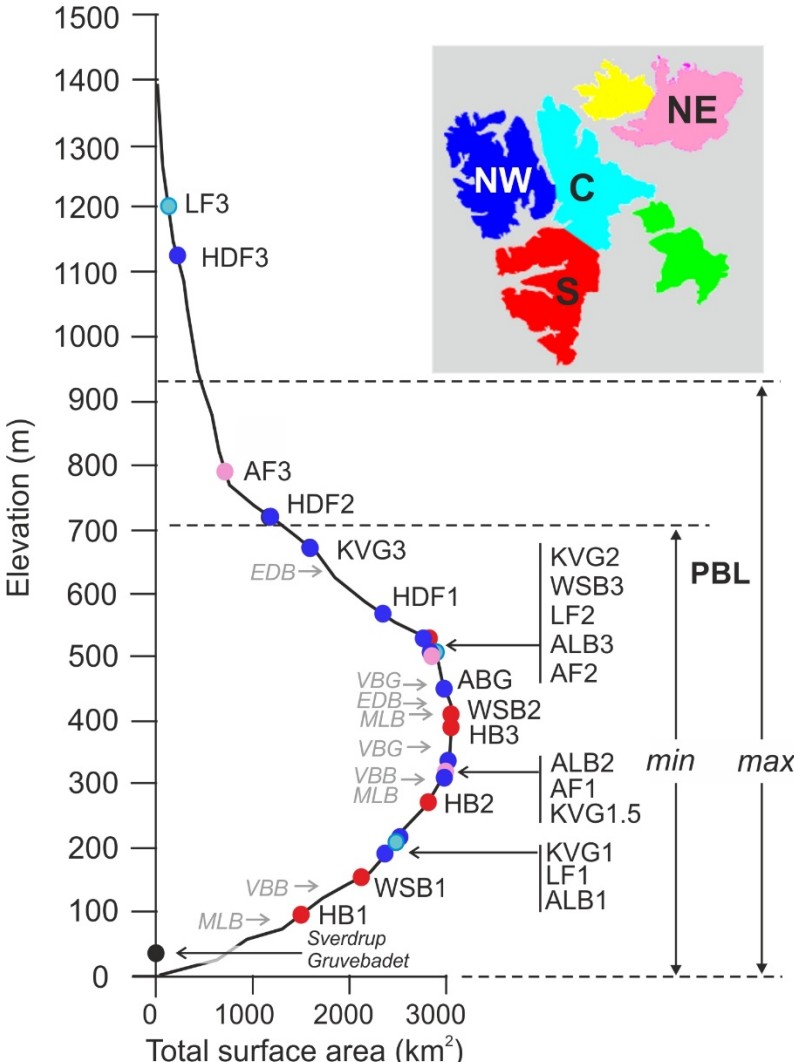

**Fig. 2.** Snow sampling sites visited during the Spring 2016 glacier survey, plotted with respect to elevation and glacier surface hypsometry (black line; from James et al., 2012). Each circle corresponds to sampling location, and those on glaciers are labelled as in **Table 2**, and color-coded per region (inset, from König et al., 2014). The grey arrows and symbols to the left of the hypsometric curve indicate elevations at which surface snow samples were collected during the Spring 2017 survey. Also shown is the minimum (winter) and maximum (summer) thickness of the Planetary Boundary Layer (PBL) in the maritime sector of the European Arctic, based on ERA-40 reanalysis over 1969–2001 (Esau and Sorokina, 2009).

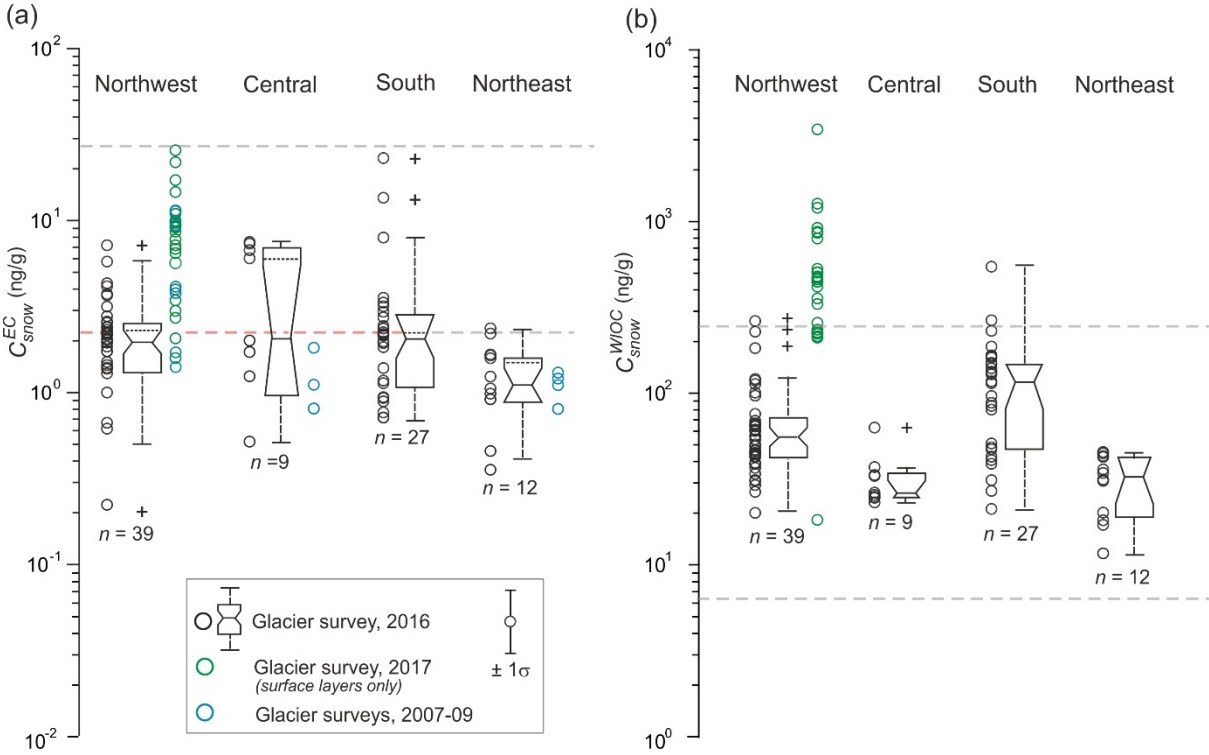

**Fig. 3.** Measurements of (a) $C_{snow}^{EC}$ and (b) $C_{snow}^{WIOC}$ on Svalbard glaciers, grouped by geographic sectors (defined on **Fig. 2**). The box-whisker plots only include measurements from snowpit samples taken on the glaciers surveyed in April 2016. Box heights give the interquartile range, and plus signs ("+") are outliers. Notches bracket the 95 % confidence limits on the median. The ± 1σ error bar in the legend box corresponds to a coefficient of variation of 40 % on individual $C_{snow}^{EC}$ or $C_{snow}^{WIOC}$ values. The dotted horizontal traits on the box plots are the estimated medians if $C_{snow}^{EC}$ values <1 ng g$^{-1}$ are excluded. Green circles are $C_{snow}^{EC}$ in surface snow layers sampled on glaciers of NW Spitsbergen in 2017 (**Table S2**) and blue circles are the median values of $C_{snow}^{EC}$ and $C_{snow}^{WIOC}$ in snowpit samples collected on glaciers in 2007–09 (Forsström et al., 2013). The grey dashed lines bracket the interquartile range of $C_{snow}^{EC}$ and $C_{snow}^{WIOC}$ measured in surface snow layers on the glacier Austre Brøggerbreen, NW Spitsbergen, between 2007 and 2018 (this study).

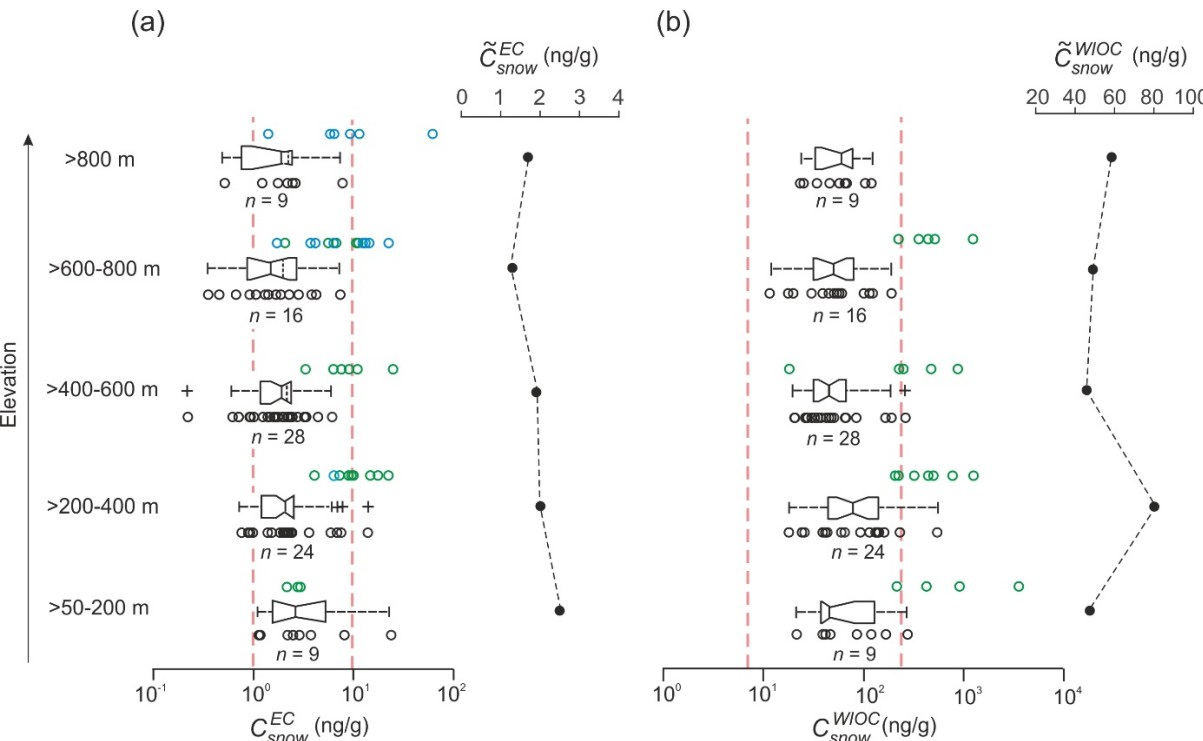

**Fig. 4.** Measurements of (a) $C_{snow}^{EC}$ and (b) $C_{snow}^{WIOC}$ on Svalbard glaciers, grouped into discrete elevation bins. Data symbols and box plots are defined as in **Fig. 3**. Altitudinal profiles of the median values of $C_{snow}^{EC}$ and $C_{snow}^{WIOC}$ ($\tilde{C}_{snow}^{EC}$, $\tilde{C}_{snow}^{WIOC}$) are shown separately on linear scales. Note that the elevations on the latter plots are ordinal only, as $\tilde{C}_{snow}^{EC}$ and $\tilde{C}_{snow}^{WIOC}$ are computed from sample data collected over a range of altitudes within each bin.

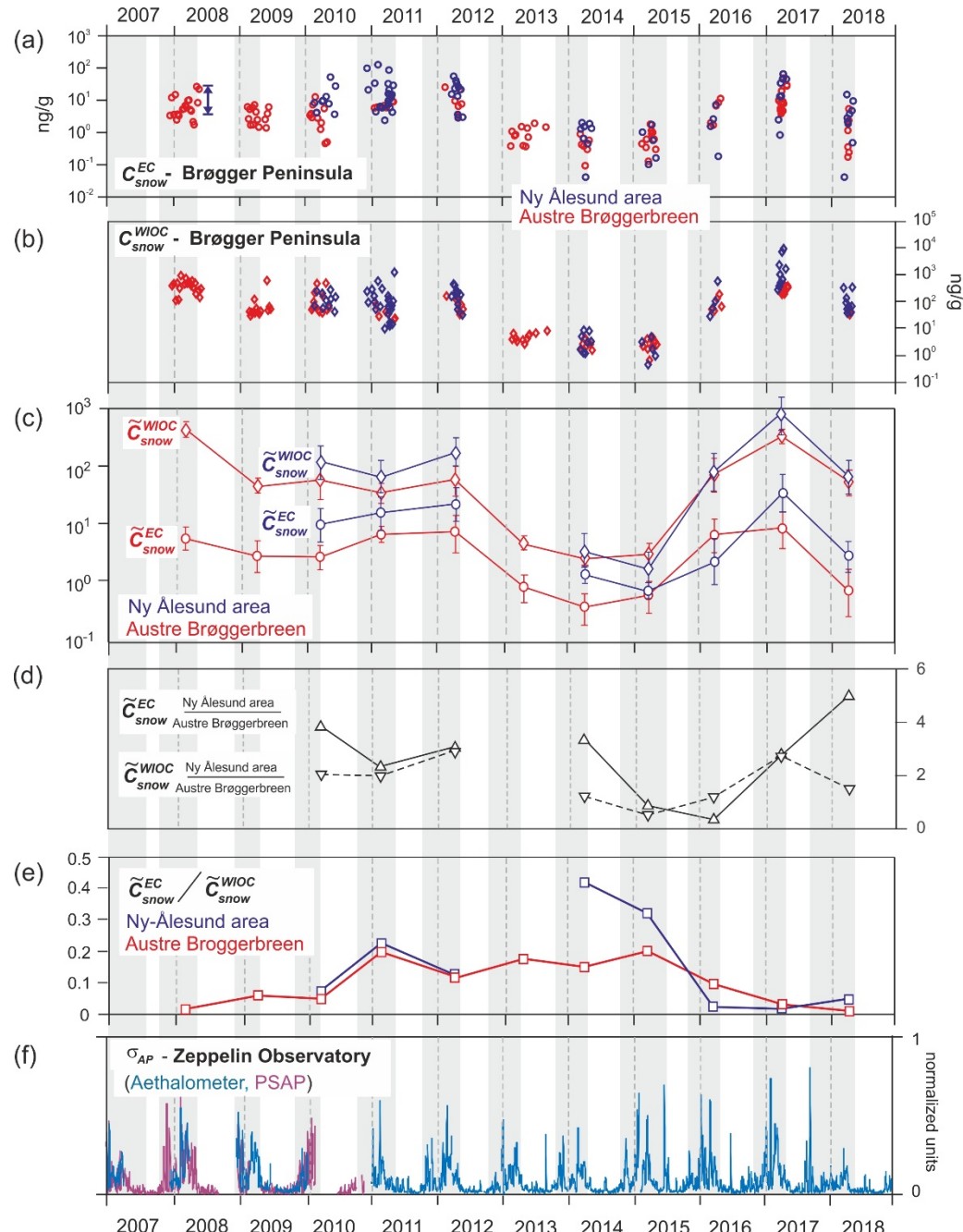

**Fig. 5.** (a) $C_{snow}^{EC}$ and (b) $C_{snow}^{WIOC}$ in surface layers on central Brøgger Peninsula, Svalbard, 2008–18. The double-headed arrow in (a) is the interquartile range of $C_{snow}^{EC}$ measured near Ny Ålesund in March 2008 by Aamaas et al. (2011). (c) Median values of these data ($\tilde{C}_{snow}^{EC}$, $\tilde{C}_{snow}^{WIOC}$) near Ny Ålesund and on Austre Brøggerbreen, with error bars representing 95 % confidence bounds (see text for details). (d) Ratio of $\tilde{C}_{snow}^{EC}$ (upright triangles) and of $\tilde{C}_{snow}^{WIOC}$ (inverted triangles) near Ny Ålesund and on Austre Brøggerbreen. (e) Ratio between $\tilde{C}_{snow}^{EC}$ and $\tilde{C}_{snow}^{WIOC}$ at these sites. (f) Weekly averages of the aerosol absorption coefficient $\sigma_{AP}$ measured at Zeppelin Observatory (Eleftheriadis et al., 2009). These data are shown here to highlight the timing of winter/spring maxima in BC aerosol mixing ratios in the study area.

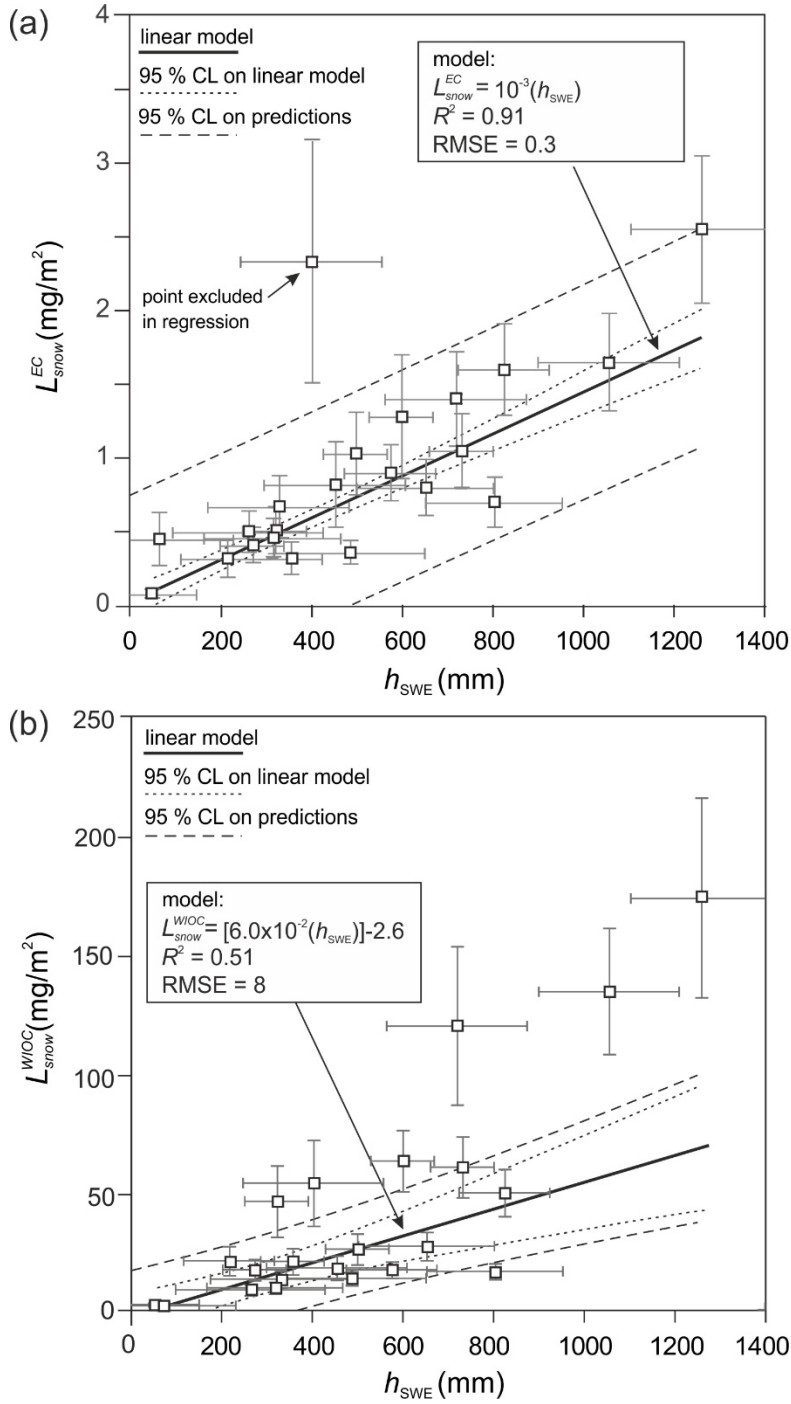

**Fig. 6.** Scatterplot of (a) $L_{snow}^{EC}$ and (b) $L_{snow}^{WIOC}$ against $h_{SWE}$ based on measurements from Svalbard glacier snowpits from the April 2016 survey. The vertical error bars are $\pm 1\sigma$, and take into account uncertainties in $h_{SWE}$, $C_{snow}^{EC}$, and $C_{snow}^{WIOC}$. The horizontal error bars ($\pm 1\sigma$) are based on the RMSE of the values of $h_{SWE}$ predicted by the snow model, relative to field observations (Table 2 in van Pelt et al. 2019). Weighted linear regression models fitted to the data are shown with 95% confidence bounds.

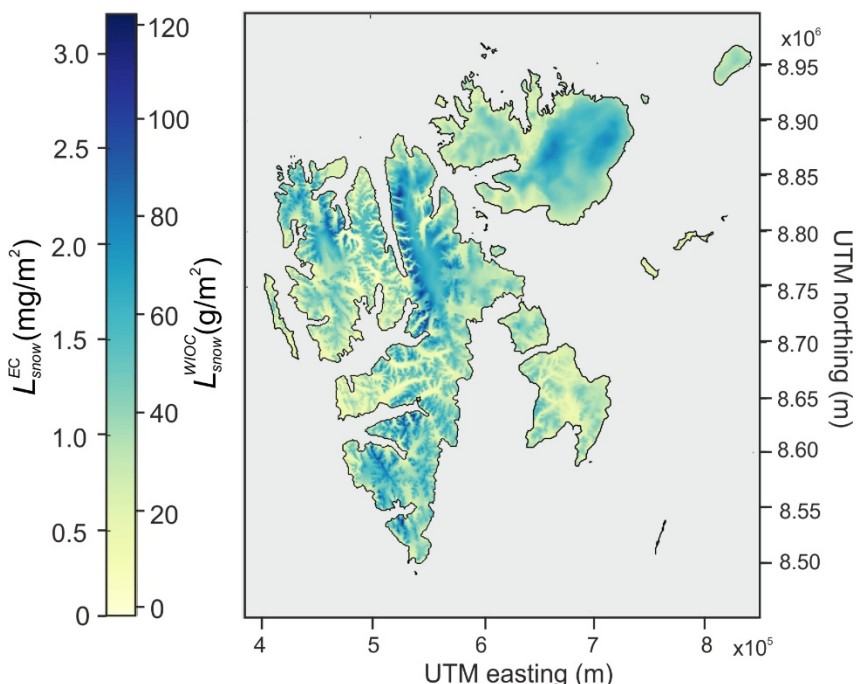

**Fig. 7.** Map of the estimated $L_{snow}^{EC}$ and $L_{snow}^{WIOC}$ in the late winter 2015−16 snowpack over Svalbard, based on the empirical relationships shown in **Fig. 6**, applied to the map of $h_{SWE}$ between 1 Sept. 2015 and 30 April 2016 generated using the snowpack model (Van Pelt et al. 2019). Note that these maps do not include EC or WIOC deposition in snow from local point sources of pollution around settlements such as Barentsburg, Longyearbyen or Ny-Ålesund.

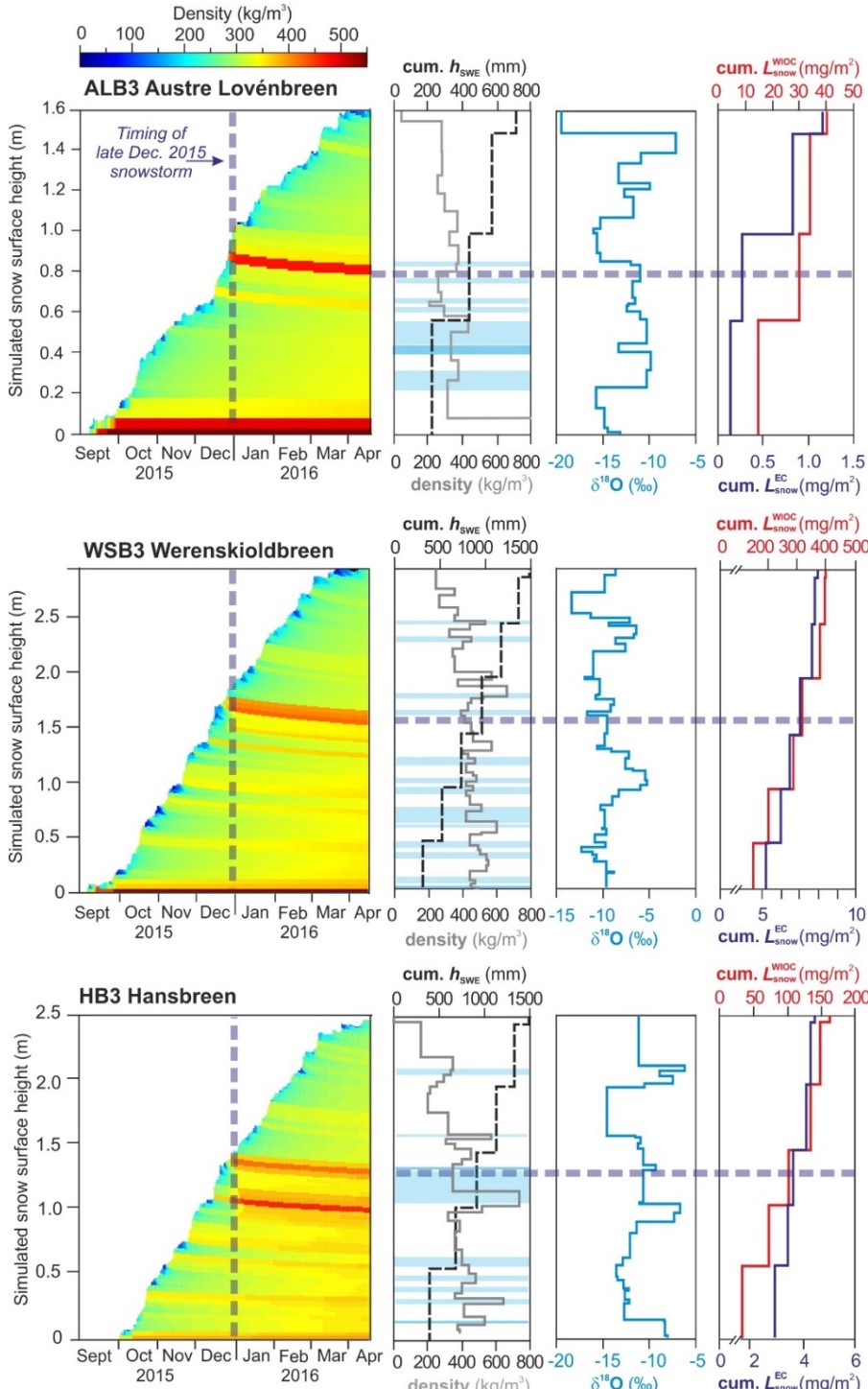

**Fig. 8.** Simulated evolution of the snowpack from Sept 2015 to April 2016 at three glacier sites on Spitsbergen, compared with measured profiles of density, cumulative $h_{SWE}$, $\delta^{18}O$, as well as cumulative $L_{snow}^{EC}$ and $L_{snow}^{WIOC}$ in the snowpack. The $h_{SWE}$ over the EC and WIOC sampling intervals was computed using the discrete snow layer density data. Where density measurements were missing, values from comparable layers in other snowpits were used. Snow layers with $C_{snow}^{EC}$ <1 ng g$^{-1}$ were assigned a value of 0.5 ng g$^{-1}$ for $L_{snow}^{EC}$ calculations. Icy snow and discrete ice layers are shown as pale and darker blue lines, respectively.

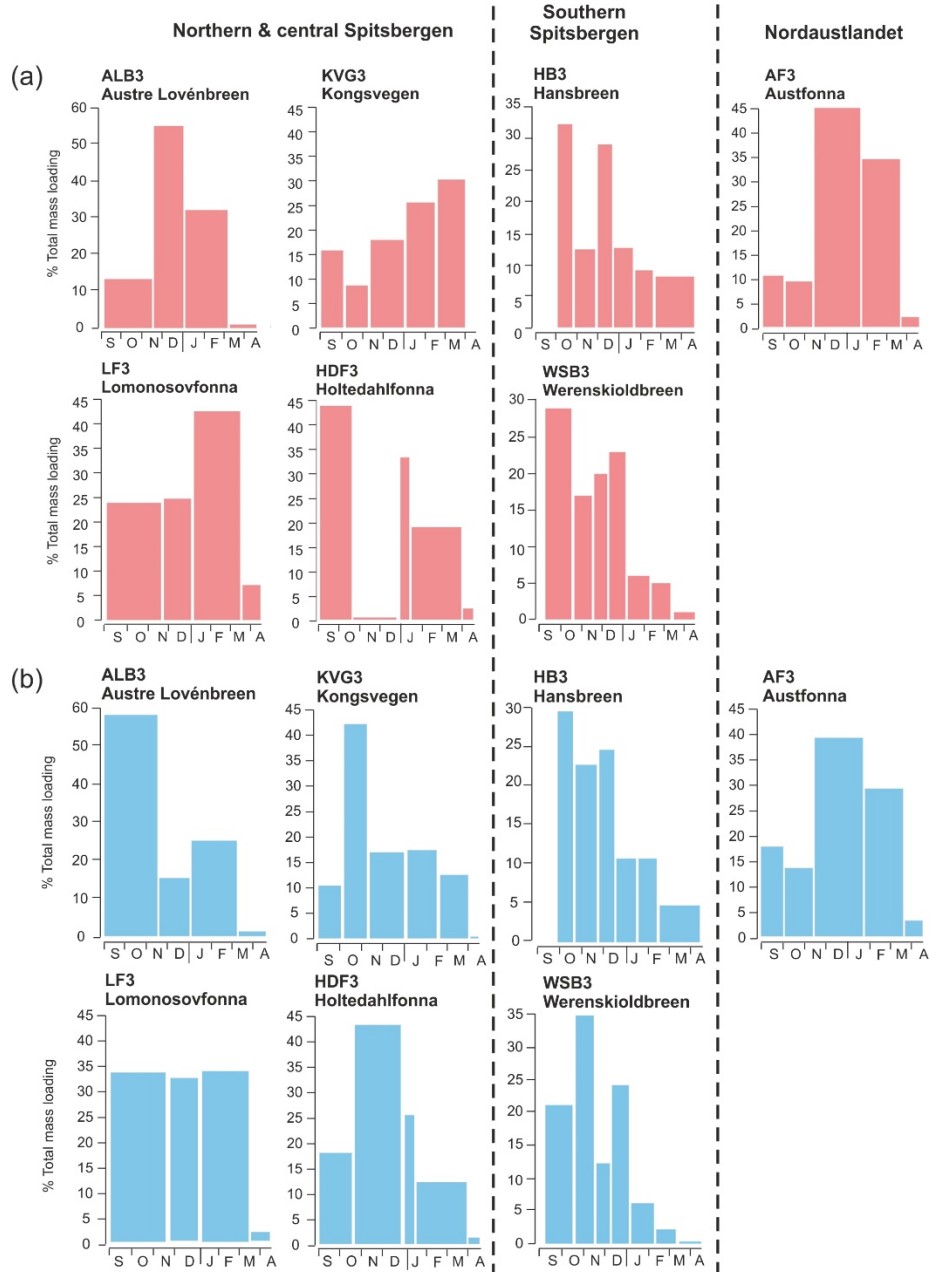

**Fig. 9.** Sub-seasonal increments of (a) $L_{snow}^{EC}$ (red) and (b) $L_{snow}^{WIOC}$ (blue) in the accumulation zone of Svalbard glaciers sampled in Spring 2016, as estimated using the snowpack model (e.g., **Fig. 8**).

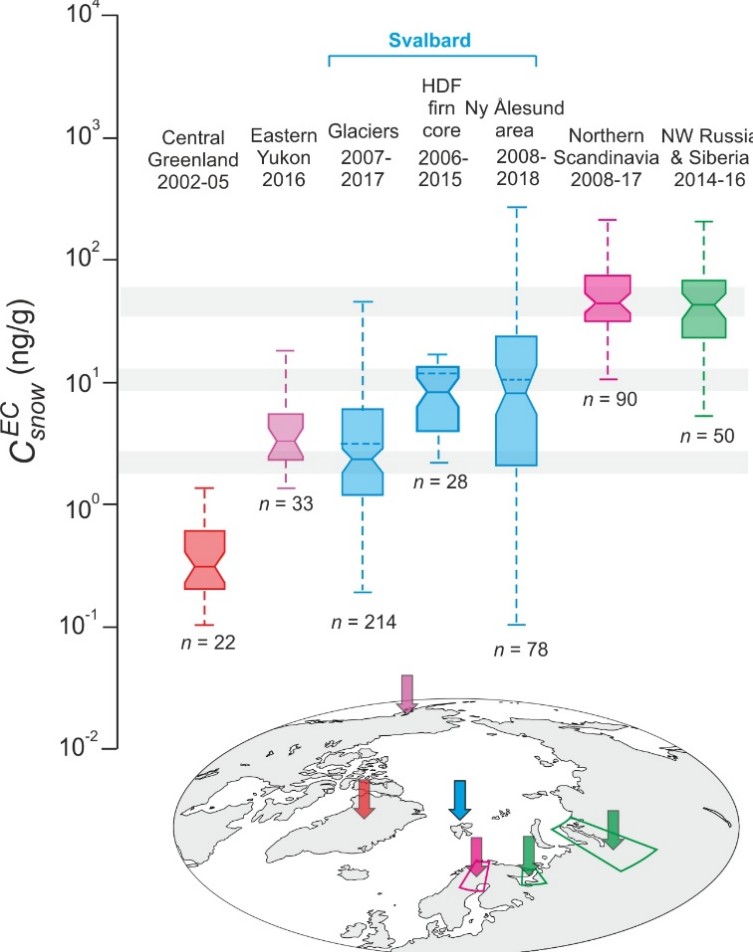

**Fig. 10.** Variations of $C_{snow}^{EC}$ in snow across circum-Arctic sites, color-coded by region. Only data obtained by thermo-optical analysis using either the NIOSH 5040 or EUSAAR_2 protocols are included (see **Table S4**). The box-whisker plots are as defined in **Fig. 3**, but outliers were removed for clarity. The plot "Glaciers 2007-2017" combines all glacier snowpack data from the present study, as well as from earlier glacier surveys by Forsström et al. (2009, 2013). The plot "HDF firn core" is based on the analysis of a firn core from Holtedahlfonna (Ruppel et al., 2017), and the plot "Ny Ålesund area" is based on the surface snow data presented in **Fig. 5** of the present study. Other data sources are: Greenland: Hagler et al. (2007); northern Scandinavia: Forsström et al. (2013), Meinander et al. (2013), Svensson et al. (2013, 2018), and unpublished data (**Table S5**); Russia and Siberia: Evangeliou et al. (2018); Yukon and Sweden: unpublished data (**Table S5**). Data from Greenland and the Yukon span 3–6 years of accumulation in snow, while the Holtedahlfonna firn core spans an estimated ~8 years (2006–14). The shaded grey bars indicate, for different median median values of $C_{snow}^{EC}$, the estimated spread (interquartile range) of results that might arise from methological differences in the TOT analyses between studies.