# Peer review of "Elemental and water-insoluble organic carbon in Svalbard snow: A synthesis of observations during 2007–2018."

_Atmospheric Chemistry and Physics, 2020_

## Referee Comment (RC1) · Anonymous Referee #1 · 24 Aug 2020

Spatiotemporal variability of elemental and organic carbon in Svalbard snow during 2007-2018

This paper presents the mass concentrations of EC and OC in snowpack collected in Svalbard using TOT technique. Authors discuss the variability of these values, especially versus altitude. There have not been such extensive data in this area, as are presented in this paper. The features shown are potentially valuable. However, there are important problems that need to be addressed in considering the publication of this paper in ACP.

Major comments

**In this study, TOT technique is used for the analysis of EC and OC in melted snow. In the present paper, it is simply stated very qualitatively that "$C^{EC}_{snow}$ in snow samples that contained visible dust may be underestimated". Although potential errors are discussed in some detail in the Supplement, no quantitative estimate of the uncertainty of the data could be given after all.**
Generally, the application of TOT technique to snow samples is not well established. Characteristics of the problems associated with TOT method for airborne particles may not be the same as those for the measurements of EC and OC in water samples. Authors cite the previous study of Lim et al. (2014), which compared EC concentrations with those of BC measured by an SP2 instrument. The methodology used by Lim et al. is quite straightforward and supported. However, it was made for a limited number of samples and it is difficult to generalize their results. It is a responsibility of the authors of this paper to make similar inter-comparisons using their snow samples.
  It is very difficult to understand to what extent the observed features presented in this paper are natural phenomena or due to measurement errors. The methodology used in this paper lacks sound scientific basis, considering that the presentation of the absolute values of EC and OC is the central point.
Therefore, much more detailed analysis of the TOT technique must be made. The situation is worse for section 4.3, as described in specific comments. If there are no substantial improvements, I recommend rejection of this paper for publication in ACP, although there seems to be important findings and implications derived from the data.

Below, I have given specific comments.

**pages 6-7.**
Snowpack model is used for this study. It is desirable to give uncertainties of the model estimates.

**Page 7**
$C^{eBC}_{air}$ is used for some data interpretation. Here, no critical evaluation of this data is given. Single parameter MAC is used to derive this quantity. However, no basis is given to show that this methodology is supported. In addition, different instruments are used to derive this quantity. Associated uncertainties should be given.

Page 8.
Table 2. It will be useful to include average altitudes for these sites to show altitude dependence of the quantities given in this table.
Table 2 and Figure 5 will be most important in this paper. It will be useful if these

locations are clearly shown in Figure 1, for example. There are a number of snow sampling locations given in Figure 1. It is difficult to relate them with Table 2 and Figure 5 for readers unfamiliar with the geography of Svalbard. You may consider using abbreviations, such as used in van Pelt et al. (2019).

**Page 10, L298-302 and related discussion.**
$L^{EC}_{snow}$ is almost proportional to $h_{SWE}$ up to $h_{SWE} = 800$ mm. Apparently, the relation becomes non-linear due to the two data points at $h_{SWE} > 1200$ mm. There is no physical basis to fit the data by an exponential function. In the linear region of $h_{SWE} < 800$ mm, the slope gives an average $C^{EC}_{snow}$ for this data set. I consider that $C^{EC}_{snow}$ for $h_{SWE} < 800$ mm is constant as a first approximation, especially considering that the uncertainty is not well defined. It is likely this slope agrees with the average $C^{EC}_{snow}$ of individual sample values and this should be mentioned.

**Section 4.1, page 9-10.**
Spatial and temporal variabilities of EC and OC are discussed here. It is difficult to follow them in a short time due to many numbers presented here. This part should be re-structured, may be by using a figure or table. Using abbreviations may also help. Details should be moved to Supplement.

**Section 4.1, page 12, L354-365 and page 15, L 460**
Two data points at $h_{SWE} > 1200$ mm are interpreted as the result of inflow of polluted air from south associated with large cyclonic storms. Only $L^{EC}_{snow}$ is discussed here, but it is the linear slopes of these points from the origin ($C^{EC}_{snow}$), that are most relevant here. It will be useful to check if $C^{eBC}_{air}$ at Zeppelin showed large increases during these events. It should be enhanced by large factors if air transported from south was polluted. It is stated that storms and landfall caused large $L^{EC}_{snow}$. It may explain large $h_{SWE}$, but it will not explain large $C^{EC}_{snow}$. The effects of storms on $h_{SWE}$ and $C^{EC}_{snow}$ should not be mixed.

**Section 4.2 and other parts**
Variation of $C^{EC}_{snow}$ with altitude is an important point, again depending on the reliability of the data. Considering its importance, more detailed discussion on possible causes of this feature should be given.

**Section 4.2 and other parts**
The effect of dry deposition is considered to be small from the analysis Figure 6. I agree with this analysis. However, the effect of dry deposition may be substantial for other snow samples. More considerations should be made on this point.

**Section 4.2, page 13, L 398-427**
The analysis made here assumes that $C^{eBC}_{air}$ at the surface is directly linked with $C^{BC}_{snow}$. No detailed analysis is made to support this assumption and the discussion based on this assumption is very weak. It does not add solid materials to this paper. Therefore, it should be delated.
A minor comment. $W$ is not non-dimensional. A proper unit should be given to it.

**Section 4.3.**
In this section, $C^{EC}_{snow}$ data from different groups using different TOT methods are collected. There are two problems here. In this part, no detailed analysis and discussion are made on the uncertainties of EC data by different groups. It is not scientifically

sound to simply collect the data without any critical evaluations.

A second problem is that this section has little relevance with the major points of this paper and deviates from the mainstream of this paper. The circum-Arctic data are best analyzed and discussed in detail as a separate paper. Therefore, this section should be deleted.

Minor comments

Numerical values of parameters such as $C^{EC}_{snow}$, $C^{OC}_{snow}$, $L^{EC}_{snow}$, and $L^{OC}_{snow}$ are given in 3-5 digits at many places, including abstract. Considering that the uncertainties cannot be given and are potentially large, these numbers should be given in 2 or 3 digits.

**Page 2, L45**

BrC is referred to here. But no discussion is made in the discussion of OC in this paper.

**Page 14, Line 413**

"f" is not clearly defined. Is it zero or one ?

**Page 16, Line 484**

UL may be UW.

---

## Referee Comment (RC2) · Anonymous Referee #2 · 1 Sep 2020

Reviewer ID# Manuscript No.: ACP-2020-491 Authors: Christian Zdanowicz et al. The title of manuscript: Spatiotemporal variability of elemental and organic carbon in Svalbard snow during 2007-2018

General comments:

This paper is very informative and integrative. It includes a large dataset covering spatial distribution of concentrations ($C_{snow}$) and column loading mass ($L_{snow}$) for both EC and OC (water insoluble OC only) across Svalbard archipelago as well as temporal variation of $C_{snow}$ over the period of more than 10 years (2007-2018) from central BrØgger Peninsula, involving 49 sites and 324 samples. Those data were used

together with meteorological and aerosol measurements in combination with snowpack modeling. The variation of EC and OC deposition in Svalbard snow across latitude, longitude, elevation and time were investigated. The averages of snowpack loading of EC and OC across Svalbard for the winter of 2015-2016 were estimated. A range of possible snow scavenging rates of EC were obtained by comparing the estimated Csnow (via Cair) with the measured Csnow of EC. Overall, no spatial and temporal trends are detected. The authors dedicated lots of effort to this work and is worthy to be published. This work would make more data available in the arctic region (although with large uncertainties) and facilitate the arctic climate research. However, I have some concerns as follows:

- The weakest aspect of the work is the methodology itself, which could not be avoided likely. Thus, it weaken the impact of the work in general. To ensure a methodology able to archive the expected objective, standards/references should be run using the same procedure before any ambient samples is measured. Although there is no consensus on BC standard/reference, at least, some proxy of standards (e.g., graphite or fullerene soot or NIST SRM 1649a for EC; whereas water insoluble organic carbon are easy to find...) could be utilized. However, no information available in the paper as well as in the cited references regarding using standards/references in the filtration process, transferring BC particles from the melting snow water onto filters.

- The title should be modified since not the total organic carbon (OC) instead of only water insoluble OC contents were measured here (without knowing the filtration recovery rate). Authors should clarify the definition of OC through the entire paper accordingly.

- Due to many possible sources of errors in these snow sample processing and analyzing, it is needed including more error-analysis to get overall uncertainties properly, including but not limiting to filtration loss for both EC as well as water solvable OC, underestimated EC due to Fe3O4 (reduction-oxidation chemistry under 500 degree C) from dust & overestimated EC and OC due to carbonate etc.). As stated in Summary and Conclusion, no discernible spatial and temporal trends as well as patterns according elevation are obtained. It is not sure to what extent these statements are true or due to the large uncertainties caused by the methodology itself.

- In addition, authors include too many contents in one paper to keep the logic flow clearly and concisely and to link them via a main theme harmonically. It is noticed that due to a large dataset covering different spatial, temporal and discipline, too many names of site/glacier and terminology are introduced, making it hard to follow the paper through. For examples, 49 sites are indicated in Table 2, but only 19 sites on Fig. 1. (map). What are the relationship between Table S1, S2 & Fig. S2 (map)? Why are all site descriptions not integrated within one table and one map, clearly arranging them into three categories as follows; 1). 2015-2016: 7 glacier snowpack campaigns (22 sites) 2). 2008-2018: Brogger Peninsula surface snow (3 sites) 3). 2016-2017: 6 glaciers Northwestern Spitsbergen ( ? sites) The descriptions in section 2.1 "Field sampling" are not consistent with the descriptions on Table 2 for the 4 categories! Very confusing. . .

Hope these issues and concerns be addressed before the publication.

Specific comments:

L42: While it's importance is realized for climate forcing, it is understood that BrC is the kind of organic carbon, including water-soluble organic carbon (i.e., WSOC), mostly polar organic compounds (e.g., oxygenated OC or organic nitrogen compounds or organosulfur compounds). However, the OC in this manuscript does not include the fraction of WSOC, which was lost by $\sim 80\%$ through the filtration process. Please re-write this part accordingly and relate BrC to the water insoluble OC as WIOC (e.g., solvent soluble OC are also part of BrC).

L60-65: There are some confusion here since this study followed the filtration method by Forsstrom et al (2009, 2013) and utilized the EC/OC analysis by EUSAAR_2 but providing the citation by Chow et al, 2004. Please clarify.

L66: "OC" should be defined and clarified here

L70-71: I would prefer "scavenging ratio" to "washout ratio".

L75: it would be better replacing the section title with "2.1.1 April 2016 glacier survey"

L84: Fig. 2 is hard to follow; particularly for the caption, (the color in lower panel does not match these in upper panel). Please modify the figure to make it easier to understand or take it out (which not belongs to the key figures).

L87: please replace "stable oxygen isotope ratio (d18O)" with "stable oxygen isotope ratio (d18O) in water"

L104: Please spell out "ERA" for the first time.

L111: Suggest modifying the section title to "Surface snow monitoring (2007-2018), Brogger Peninsula"

L125: These additional samples in Table S2 are confusing as the sampling sites and the date are partially overlapped with these samples in Table S1. Are there any special purpose for those samples in Table S2? If not, it make more sense to include them in section 2.1.1 and change the current title to "2.1.1 2016-2017 glacier survey". Suggest to make 2 sub-sections: one is about the major survey in April 2016 and the other is about the additional irregularly sampling by NPI staff (2016-2017).

L130-140: As for EC and OC analysis of the snow samples, there are several steps involved, including snow processing (melting & filtration of particles, i.e., EC & WIOC on to filters) and EC/OC analysis. Large amount uncertainties would associated with these steps and the uncertainties of the snow processing are unknown largely. Author stated following the procedure by Forsstrom et al., 2009. However, the procedure by Forsstrom et al., 2009 was only for EC and the procedure regarding how to deal with the WIOC was not described, which is very important to this study. Author should provide more details of description for the snow processing steps, e.g., how many minutes were used for melting snow? and what kind of device was used for filtration of snow ?

and how many minutes were used for the filtration? It is suggested to use some proxy of reference EC and WIOC (mentioned above) and ionized water to get the recovery rates for the filtration process.

L 136: The definition of OC mass measured (i.e., WIOC) should be clarified here also.

L141-153: It is suggested that more detailed error analysis is added here, as mentioned in general comments.

L67: Please modify the section title to "2.3 d18O analysis in snow water"

L168: Please replace the expression of "The stable isotope ratio of oxygen (16O:18O)" with "... (18O:16O)".

L167-172: A sentence should be included here regarding why the authors should include d18O data in the study (Fig. 8). Otherwise, the d18O should be moved to "supplementary materials", together with all support data. Thus, the main theme could be better presented without detraction from those supporting data.

L173-L216: It is suggested to move the content in section 2.4 to "supplementary materials" to make the theme stick out.

L219: Please use plain language to explain "skewness" as a statistic concept. What does it mean applying to your data?

L218-L230: This paragraph is full of numbers and hard to follow what the author would like to express. It is suggested either totally deleting this paragraph or focusing on the description of data structure including Skewness and LOD and how would the data structure affect the interpretation of this dataset and the comparison it with other studies.

L231: L75: Please modify the current title to "3.1 April 2016 glacier survey", being consistent with 2.1.1.

L261-270: It is suggested to include all the annual median concentration data (i.e.,

Csnow on EC and OC) from both Ny-Alesund and Austre Broggerbreen shown on Fig. 5c in a table, in order to better understand the paragraph.

L272: Ensure that the section title be consistent with 2.1.1 and 3.1

L273: the general statement for "no discernible zonal or latitudinal gradient of Csnow on EC or OC across Svalbard" is not convincing, as the uncertainties are not well investigated.

L278-279: highest among what ? This sentence is not clearly expressed. In fact, there is not much difference shown on Fig. 3 and 4.

L282-283: This statement seems not supporting by the data shown in Figs, S4-S5, and Figs S8-S9. As suggested, it is better to table all the data in the Supplementary Material to see clearly. L299-L302: Please check the numbers mentioned in L299-L302, which are not consistent with those in Fig 6a and 6b.

L272-365: The Section 4.1 is too long to follow, covering the discussion on different topics, including - surface Csnow of EC and OC - Lsnow on EC & OC and the relationship between Lsnow and hSWE - Snowpack modeling and simulated hSWE, and Lsnow (via extrapolating the relations shown in Fig. 6) - Simulated total EC and OC accumulated mass during the winter of 2015-2016 (Sept. 2015-April 2016?) across Svalbard and derived monthly and daily deposition rate (i.e., mg C m-2 mo-1 or mg C m-2 day-1) in the area. - Simulated snowpack profiles and related them to the cumulated mass of EC and OC (i.e., Lsnow of EC and OC) shown in Fig. 8 and Fig. 9 and discussion snow dynamics impacts.

I would suggest to re-arrange those contents in one section 4.1 into several sub-sections as 4.1.1, 4.1.2, 4.2.3 etc. accordingly based on topics mentioned above. Author may consider moving all snowpack modeling & snow dynamics related contents to the Supplementary Materials and briefly include the most relevant results here accordingly to make the main theme clear.

[Figure]

L366 -427: Similar to section 4.1, suggest to re-arrange this section 4.2 into two sub-sections as: - L367-397: 4.2.1 "Temporal variation of Csnow on EC and OC" - L398-427: 4.2.2 "Scavenging rate of EC, Brogger Peninsula"

L395-397: It is suggested to begin with the sentence via using "To understand the possible role snowfall anomalies over center Brogger Peninsula..." instead of "To control for the possible role of snowfall rate..."

L429: Suggested to modify the section title as "Comparison of Csnow EC in Pan-arctic perspective".

P34, Fig. 9 All the sites labelled in the plots here should be included in a map identified each of them. ALB3 could not found in Fig. 1.

In Supplementary Section: Under "EC and OC analyses: Additional information": Uncertainties estimation and analysis should be paid more attention and include more discussion as mentioned in the General Comments. In addition to the uncertainties of sigma_sh, sigma_u and sigma_f, for EC, there are sigma_ue (Under Estimate) due to dust (e.g., Fe2O3) and sigma_oe (Over Estimate) due to carbonate; whereas for OC, there are sigma_lwsoc , i.e., loss of WSOC ($\sim$ > 80% of total OC, Hagler et al., 2007) during filtration processing, and sigma_oe due to dust and carbonate. The overall uncertainties should be derived via error propagation.

Fig. S2: Please show the relative location of Fig. S2 to Fig. 1.

Fig. S4-S5: Since the log scale is used, it looks no much variation observed. In fact, large variation may exists. It is suggested to list all the data in a table,

Please also note the supplement to this comment:
https://acp.copernicus.org/preprints/acp-2020-491/acp-2020-491-RC2-supplement.pdf

---

## Author Comment (AC1) · 3 Nov 2020

**acp-2020-491**
**Spatiotemporal variability of elemental and organic carbon in Svalbard snow during 2007-2018**

**Author's response**

Our response is organized as follows: We start with a general discussion of the main criticisms raised by the reviewers, and of aspects of the analytical method. Some parts of this discussion have been added to the revised manuscript or its supplement. We follow by a point-by-point reply to the more specific criticisms and/or suggestions made. To avoid needlessly repeating arguments when we respond to detailed comments, we will refer to the appropriate paragraphs in the discussion (numbered in brackets below). A list of the major changes made to the manuscript is provided at the end of this document.

**General discussion** *(reply to major comments)*

[1] Both reviewers criticized the paper for providing an insufficient discussion and analysis of the uncertainties associated with the method and protocol used (thermo-optical transmission, TOT). One reason for this is that discussions of these uncertainties had already been provided in previous papers, and we did not want to repeat these discussions (see for example sections 2.1 in Aaamas et al., 2011; 2.2 in Svensson et al., 2013; 3.2. in Forsström et al., 2013, and 4.4. in Forsström et al., 2013). We have now tried, below and in the revised manuscript, to provide clarifications. Note that the figure and table numbers cited below are those in the original manuscript submitted to ACP, unless otherwise specified.

[2] Analytical methods are of course constantly being revised, re-evaluated and improved. The fact that there remain unquantified uncertainties in the TOT method when applied to snow and ice does not imply, as reviewer 1 puts it, that *"the methodology used in this paper lacks sound scientific basis"*. This is overstated, considering the large body of scientific literature that has already been dedicated to the development of this method over the past 30 years, and the many studies published since the 1980s that applied the technique to quantify EC and OC in snow both in polar settings (e.g., those listed in our Fig. 9) and in other, non-polar settings (e.g., Himalayas, Andes, Japan). In our paper, we have tried to quantify those uncertainties in our methods that are known, using our own data, or data from previous, recent studies that are especially relevant (those using comparable or identical methods). If future studies of the TOT method uncertainties (e.g., effect of dust) show that the results presented here are questionable, then they will be corrected or superseded, as is the normal course of scientific research.

[3] The main issue with accuracy and reproducibility of EC quantification in aerosols or snow by the TOT method, which plagued early published studies, was the inconsistent use of different thermal protocols, for e.g., using different EC-OC temperature split points (A comparable issue with the SP2 method is the inconsistent use of different aerosolization techniques with different size-dependent efficiencies). Other sources of uncertainty are discussed further below. In our study, we used the standardized EUSAAR2 protocol, which was developed to minimize some artefacts that can affect EC determinations in both aerosols or snow-filtered particles, for e.g. $CO_2$ evolution from carbonatic carbon (CC) and in situ charring. The EUSAAR2 was introduced by Cavalli et al. in 2010. Snow samples collected near Ny-Ålesund in 2007 ($n = 2$) and in 2008 and 2009 ($n = 18$ and 14, respectively) had been analyzed with the earlier NIOSH-5040 protocol (Birch, 2003). Later intercomparison of the two protocols revealed that NIOSH-5040 yielded EC concentrations that were about half of those obtained by EUSAAR2, and a correction factor was therefore applied to those earlier measurements, as was done in Forsström *et al.* (2013). *[Note: These details were missing from the methods in the manuscript, and have now been added].* All other snow samples in this study were analyzed with the EUSAAR2 protocol, and using the same instrument, to maintain consistency and ensure comparability with earlier surveys reported in Forsström *et al.* (2009, 2013).

[4] Reviewer 2 expressed concern that no measurements of standards were presented in the manuscript. This is partly on account of the nature of the method, which is based on filtered particulates. It is indeed very challenging to prepare filters with pre-determined masses of BC particles that share characteristics of real aerosols, and with loadings (mass densities) comparable to those found in Arctic snow.

[5] However it does not mean that TOT measurements are unconstrained by *any* comparisons with standards. We refer the reviewers to the recent work by Svensson et al. (2018), cited in our manuscript. In this study, 36 filters were prepared from sonicated, water-ethanol suspensions containing different known amounts of NIST-2975 diesel soot, or of chimney soot. The volumes of soot suspensions were chosen so as produce particle mass densities on filters comparable to those typically obtained from alpine and Arctic snow samples. It was not possible to directly measure the mass of filtered particles on the filters gravimetrically, as it was too small. However, the expected mass densities could be estimated indirectly from the filtered volumes, and so it was possible to verify if the amounts of EC detected increased proportionally with the estimated filtered soot mass. All filters were analyzed with a Sunset ECOC analyzer (identical to the one used on our study) and with the same thermal protocol (EUSAAR2). The EC amounts were then compared with measurements of light absorption made on the same filters using a particle soot absorption photometer (PSAP). The results showed direct, positive linear relationships between EC mass densities and the optical depth ($\tau$) of the filters (Svensson et al., 2018, Fig. 3; $R^2 = 0.83$ and 0.92 for chimney and diesel soot, respectively), with both EC and $\tau$ increasing with the amount of soot in the filtered suspensions (i.e., decreasing with the amount of dilution). These experiments were made both at Stockholm University and at the Finnish Meteorological Institute on identical instruments, and yielded closely comparable slopes for the EC-$\tau$ relationships when using the same standard materials. Hence, there is clear evidence, based on the analysis of standard soot materials, that the mass density of EC measured on filters by the TOT method (and using the same protocols as in our study) actually produce coherent, predictable and repeatable results that are are corroborated by an independent method (PSAP).

[6] Both reviewers also raised concerns that the uncertainties in the TOT method limit our ability to confidently identify spatial or temporal patterns of $C_{snow}^{EC}$ and $C_{snow}^{WIOC}$ in Svalbard snow, thus weakening the reliability of the main findings. We are confident that this is not the case, at least for $C_{snow}^{EC}$, and explain why below.

[7] In the manuscript, we first examine how concentrations of $C_{snow}^{EC}$ and $C_{snow}^{WIOC}$ measured in the snowpack on the glaciers that were sampled in 2016 vary across Svalbard, in terms of latitude/longitude (Fig. 3) and altitude (Fig. 4). We then compare interannual variations in $C_{snow}^{EC}$ and $C_{snow}^{WIOC}$ in surface snow at two sites (Ny-Ålesund, Austre Brøggerbreen) over an 11-year period (Fig. 5c and d). In all cases, what we are comparing are the *medians of groups of measurements* (samples), not individual values. The question therefore is: Are uncertainties on the medians that arise from the methods larger than the uncertainties that arise from the overall spread of values and differences in sample sizes ?

[8] As previously discussed in Forsström et al. (2013) and in Svensson et al. (2013, 2018), the main sources of uncertainty in the TOT measurements of EC and WIOC that are not due specifically to the particle composition are: possible filtration undercatch, uneven loading of particles on the filters, and the precision (repeatability of measurements) of the Sunset ECOC analyzer itself. The effect of particle retention in containers during snow melting and filtration was previously evaluated for our filtration methodology and was found to be comparatively very small (Forsström et al., 2009; on this, more below). There is also the natural spatial variability in the distribution of EC and WIOC particles in snow, which is scale-dependent. The meter-scale is the relevant one for snowpit or surface snow samples collected in our study, and the variability of $C_{snow}^{EC}$ at this scale was previously estimated by Forsström et al. (2009) and by Svensson et al. (2013) in Arctic snow.

[9] Together, the aforementioned sources of errors combine to produce much or most of the variations between EC and WIOC mass densities measured on separate punches obtained from the same snow sample filters. When the uncertainties arising from the method and from meter-scale variability in snow are combined (by quadrature), they yield a median CV of ± 40 % for EC, as was reported in the manuscript. We assumed a similar range of variability for WIOC, in the absence of more specific information to support such estimates.

[10] To find out how these combined uncertainties affect the estimated medians for specific groups of EC and WIOC measurements, we used a Monte Carlo approach in which surrogate data were generated from the measured $C_{snow}^{EC}$ and $C_{snow}^{WIOC}$, assuming that each of the latter have normal distributions of errors with a CV of ± 40 %, as discussed above. We also included in this simulation the effect of randomly-varying undercatch of particles during filtration (median 22 %, up to 35 %), see below). The confidence intervals on the medians of data groups were then determined with, and without, including these uncertainties. An example of the results is shown graphically below.

[Figure]

[11] The figure above shows box-whisker plots of the distribution of $C_{snow}^{EC}$ measured in Svalbard glaciers, as on Fig. 3 of the manuscript. This witdth of the notches on the box-plots are the uncertainties (95 % confidence bounds) on the medians of each group of observations that arise from their overall spread and the sample size. Differences between group medians were not considered as significant if their confidence bounds overlapped. The smaller grey box plots show the spread in the estimates of the group medians that arise from the methodological uncertainties alone, as determined by the Monte Carlo approach. This is the spread of results one would expect in 500 replicate analyses of the same set of individual snow samples, assuming each observation is subject to a C.V. of 40 % due the combined effects of instrumental errors, uneven loading of particles on filters, etc. The height of the boxes on these grey boxplots is the interquartile range, so twice this width corresponds to 95 % confidence bounds on the median, assuming a normal probability distribution.

[12] What the plot above show is that in samples for which the spread of individual measurements of $C_{snow}^{EC}$ is relatively small, for example in the sample from northeastern Spitsbergen, the 95 % confidence bounds on the median EC value that arises from methodological uncertainties is about as large as that which results from the variability within the sample. In other samples, such as that from central Spitsbergen, the variability of measured $C_{snow}^{EC}$ values is much larger, and the 95 % confidence bounds on the estimated median $C_{snow}^{EC}$ are consequently far greater than those that arise from methodological uncertainties.

[13] What this implies is that at present, our ability to detect spatial differences in the median $C_{snow}^{EC}$ concentrations across Svalbard glaciers is primarily limited by the inherent large variability of $C_{snow}^{EC}$ in the snow, more than by the methodological uncertainties. One source of uncertainty that is not accounted for in the grey box plots shown above is filter undercatch. When this is factored in, the estimates of the group medians are all slightly lower (dashed red lines on the plots above), but the spread of the uncertainties for each group medians remains the same, as do their relationships between groups. We performed the same Monte Carlo exercise for the comparison of both $C_{snow}^{EC}$ and $C_{snow}^{WIOC}$ distributions in glaciers grouped by altitude bins (Fig. 4 of the manuscript), and for the interannual variations in the median $C_{snow}^{EC}$ and $C_{snow}^{WIOC}$ concentrations of snow near Ny Ålesund (Fig. 5 of the manuscript), and obtained the same results.

[14] Consequently, the main conclusion presented in our manuscript concerning spatial patterns (at least with regards to EC) remain unchanged: *"The April 2016 survey showed no discernible zonal or latitudinal gradient (...) across Svalbard"*. Note that the word "discernible" was used in the manuscript intentionally, to stress the fact that the conclusion is conditional on our capability to discern such trends amidst the variability of the data. Our analysis of method-related uncertainties (above) shows that reducing them, while obviously desirable, would not automatically increase our ability to detect spatial patterns (if they exist), given the large variability in $C_{snow}^{EC}$. Repeated surveys with larger sample sizes, however, most likely would.

[15] Furthermore, when one considers the year-to-year variations in the median $C_{snow}^{EC}$ and $C_{snow}^{WIOC}$ of spring surface snow shown on Fig. 5c, it should be noted that these variations are very similar (i.e., synchronous) at two locations (Ny Ålesund and Austre Brøggerbreen) that are separated by 5 km and 400 m of altitude difference. Given that the protocols used to generate these data were the same and consistent over most of the time period shown (2010 to 2018, when the EUSAAR2 protocol was used), it seems highly improbable that random variations caused by uncontrolled, method-related effects would result in the observed similarities.

On particle retention in containers:

[16] In our study, the overwhelming majority of snow samples collected near Ny Ålesund and on Austre Brøggerbreen by NPI staff between 2007-18 were processed as described in Forsström et al. (2013), i.e., the samples were transferred, still frozen, from bags to 1 L glass beakers and were then melted inside a microwave oven. In this procedure, melting 1 L of snow at 500 W of microwave power takes ~10 minutes, and the beakers can be agitated to homogenize the sample by mixing the meltwater with the unmelted snow fraction, which further speeds up melting. The water is then transferred into the glassware filtration apparatus, and the beaker walls are rinsed with Milli-Q water, which is also filtered. This procedure, however could not easily be used in the April 2016 glacier survey samples, because, in order to avoid shipping large snow volumes (which increase loss risks), samples were processed in different locations, some of which did not have dedicated clean microwave ovens. Instead, therefore, these samples were left to melt at room temperature inside their closed bags. Melting the bagged samples took ~24-36 hours, depending on the volume and density of the material. The meltwater was drained periodically and filtered as it was being produced, i.e. filtration was done in steps, as the samples melted, so the total duration of melting does not equate the time any water was left standing in the bags. In the end, the bags were rinsed with Milli-Q water and the rinse water was also filtered, and the added water volume accounted for. We acknowledge that these discrepancies in sample processing may have introduced some differences in the measurements of $C_{snow}^{EC}$ and $C_{snow}^{WIOC}$.

[17] It is known that BC particles can stick onto and be retained by glass or plastic bag walls, as was observed, among others, by Wang et al. (2012). These authors derived an empirical factor to correct for this, which was up to 1.5 for samples with < 80 ng/g of rBC (measured by the SP2 method). Doherty et al. (2013) chose to apply this method to rBC measured in Arctic snow samples. However the empirical correction of Wang et al. (2012) was devised from snow sampled in arid regions of China that contain typically large amounts of dust particles (30 to 90 % mass; ppm range), which favors aggregation with BC into a "scum" that adheres to the walls of containers. It is not clear that the same correction should be applied to thermo-optical EC determinations in Arctic snow with typicall much lower dust amounts. Futhermore, a laboratory experiment performed by Forsström et al. (2009) showed no evidence of a systematic decrease in detectable EC on filters prepared from melted snow (containing ~35 ng/g EC) stored in glass jars for storage times < 48 hours (see section 3.2. and Fig. 4 in Forsström et al., 2009). Therefore, we applied no corrections to our data for particle retention in containers.

On filtration undercatch:

[17] Attempts to estimate the filtration efficiency for BC particles were made in numerous studies, but as pointed out by Torres et al. (2014), the different methods used to make such estimations (and the standards or sample materials used) make it to difficult to reconcile the results, which range from 30 to 95 %. In our study, we used results from Svensson et al. (2013), Amaas et al. (2011) and Forsström et al. (2013) to estimate the undercatch, since these studies used the same methods as our own, and were based on double-filtration of real snow samples, rather than on "standards" which may have properties different than those of actual snow particulates. Results of the aforementioned studies gave estimates of undercatch that ranged between 18-35 %, with a median of 22 %. We chose not to apply systematic undercatch corrections to our EC and WIOC data, because variously-aged snow may contain variable amounts of particle aggregates that would likely cause variations in the filtration efficiency (see further below). This is particularly relevant for Svalbard snow, because winter thaw events there cause melting-refreezing cycles that promote particle agglutination of particles (this would actually reduce undercatch). We did, however, include the effect of filtration undercatch in the Monte Carlo simulation, as described above.

On the possible effects of mineral dust :

[18] A separate question is whether the presence of mineral dust (which we saw on various filters) could have led to an incorrect estimation the true EC and WIOC concentrations, and how this would affect the reported spatial and temporal variations across Svalbard. One difficulty is that at present, there are very few available data on mineral dust concentrations in snow in Svalbard. Relevant indications come from crustal element concentrations measured in snow and ice from glaciers in northwest Spitsbergen (Singh et al., 2015; Thomas et al., 2020) and west-central Spitsbergen (Casey, 2012). Together, these data suggest dust mass concentrations of a few 10s of ppb to a few ppm only, which are similar to those found in the Canadian Arctic (e.g., MacDonald et al., 2017).

[19] At least three separate, unrelated and competing dust-related effects may impact TOT analyses: (a) $CO_2$ released from carbonate minerals (carbonatic carbon, CC) and incorrectly detected as EC, (b) WIOC pyrolysis of OC by oxide minerals, which may lead to EC underestimation, and (c) the formation of BC-dust aggregates, which is more likely to occur in aged sub-surface snow than in relatively fresh surface snow (e.g., Wang et al., 2012; Kuchiki et al., 2015).

[20] The EUSAAR-2 protocol used in this study minimizes the effect of (a) above (effect of carbonates) by causing CC to evolve into $CO_2$ during the He-mode of TOT analysis, thus being detected as WIOC, rather than EC. Since WIOC is hundreds of times more abundant than EC, the effect on the measured WIOC concentrations is comparatively very small. Some authors have suggested pre-treatments to remove carbonates on filters (e.g., Evangeliou et al., 2018), but as discussed in Svensson et al. (2018), these treatments may actuallily raise more issues than they solve, and so they were not applied in our study. Furthermore, some snow samples that were collected early during the Norwegian Polar Institute snow monitoring program near Ny Ålesund were tested at Stockholm University for the effect that carbonate removal by acid fumigation had on EC quantification by TOT, but the resulting changes in thermograms were judged too minor to justify applying this procedure routinely.

[21] With regards to (b) above (effect of oxides), it is known that small amounts of mineral oxides are common in Arctic snow, even when no indications of dust is seen on filters prior to combustion. Lim et al. (2014) reported probable artefacts due to oxides in TOT analyses of alpine snow samples containing 1-10 ppm of dust, but did not give the magnitude of the associated errors on the measured EC and WIOC concentrations. Nor did they find any systematic correlation between dust amounts and the presence or absence of such artefacts in TOT thermograms. Hence at present, there are simply no firm grounds on which to base any error estimates or corrections. This will require dedicated research, which is outside the scope of our paper.

[22] Concerning (c) above (BC-dust aggregates), Wang et al. (2012) showed that in snow with >20 ppb EC, the formation of BC-dust aggregates can lead to underestimation of EC by the TOT method, unless samples are sonicated prior to filtration to break up the aggregates (sonication, however, may increase filration undercatch). The largest underestimation found in that study, for a single sample, was 20 % (80 ppb prior to sonication, 100 ppb after). For EC < 20 ppb (all other samples), the effect of sonication (hence, presumably, of aggregates) was negligible. In our own data, EC > 20 ppb occur in < 3 % of glacier samples, but in ~30 % of snow samples from near Ny Ålesund (Sverdrup and Gruvebadet sites). We hypothesized that the more frequent, relatively elevated EC levels in surface snow near Ny Ålesund (compared to Austre Brøggerbreen) are due to local combustion emissions. If locally-emitted dust (e.g., mobilized by road trafic in or near the town) caused under-estimation of EC in local snow due to dust-BC aggregates being formed, then the difference between the median EC in snow near Ny Ålesund and that in snow at Austre Brøggerbreen may in fact be larger than we sumrised, which would reinforce our hypothesis, not weaken it.

[23] Further insight on how the presence of mineral dust in snow might affect measurements of EC was provided by the work of Svensson et al. (2018). In this study, known amounts of powdered granite and of silicon carbide (SiC), used as a surrogates for mineral dust, were mixed with various amounts of soot standards (as described earlier) in aqueous suspension, and the filtered mixtures were then analyzed for EC mass density by the same instrument and method as in our paper, and also for light absorption ($\tau$) by PSAP.

[24] An important result which is relevant for the present discussion is that while the slope of the relationship between EC and $\tau$ varied for mixtures with different surrogate dusts, the amount of EC measured on the filters (and the resulting $\tau$) varied directly with the amount of soot in the filtered mixed suspensions in a consistent and predictable manner, irrespective of the proportion of added dust. This was also observed in actual snow samples from Arctic Finland and the Indian Himalayas, even tough these samples contained vastly different amounts and composition of mineral dust (see Svensson et al., 2018; Fig. 4 to 7). As shown on the graph below (based on the data from Svensson et al., 2018), some scatter does occur in the measured EC mass density from Himalayan snow samples, but it does not seem to be proportional to the dust content.

[Figure]

[25] If the presence of dust in snow samples caused a systematic underestimation of EC proportional to the amount of dust, we would expect to see the largest measured EC concentrations in samples with the lowest dust/EC ratios, but this is not what the data show. The observed pattern, combined with field observations, suggest, rather, that EC concentrations increase with dust content (up to some maximum value) as a result of BC-dust aggregation during snow ageing. (Note that both EC and dust concentrations in Himalayan snow are vastly superior to those typically found in High Arctic snow).

[26] In our study, the individual snow samples are not all equally aged. The surface samples collected near Ny Ålesund and on Austre Brøggerbreen between 2007-18 were obtained following an opportunistic sampling schedule and were not necessarily from freshly-fallen precipitation, while those collected in glacier snowpits in April 2016 mostly represent aged samples (sub-surface layers). Hence, we can expect variable degrees of dust-EC aggregation to occur in these samples as a function of their age, but the net effect on groups of samples is likely to be random rather than systematic, and, given the low amounts of dust in Svalbard snow, it would likely account for only a small part of the observed sample-to-sample variations that we measured.

Aaamas et al. (2011) doi:10.1111/j.1600-0889.2011.00531.x.
Birch, M. E. (2003), Method 5040, NIOSH Man. Occup. Saf. Health, 3, 1–5.
Cavalli et al. (2010) doi:10.5194/amt-3-79-2010.
Casey (2012) doi.org:10.1594/PANGAEA.773951.
Doherty et al. (2013) doi:10.5194/acp-10-11647-2010.
Evangeliou et al. (2018) doi:10.5194/acp-18-963-2018.
Forsström et al. (2009) doi:10.1029/2008JD011480.
Forsström et al. (2013) doi:10.1002/2013JD019886.
Kuchiki et al. (2015) doi:10.1002/2014JD022144.
Lim et al. (2014) doi:10.5194/amt-7-3307-2014.
MacDonald et al. (2017)
Svensson et al. (2013) doi:10.1088/1748-9326/8/3/034012.
Svensson et al. (2018) doi:10.5194/amt-11-1403-2018.
Thomas et al. (2020) doi:10.1016/j.scitotenv.2019.135264.
Torres et al. (2014) doi: 10.1080/02786826.2013.868596.
Singh et al. (2015) doi:10.1017/S0032247413000533.
Wang et al. (2012) doi:10.1080/02786826.2011.605815.

**acp-2020-491**
**Spatiotemporal variability of elemental and organic carbon in Svalbard snow during 2007-2018**

**Author's response to specific comments**

**Reviewer 1**

*# pages 6-7: Snowpack model is used for this study. It is desirable to give uncertainties of the model estimates*

Indeed it is. The best estimate we have of the accuracy of the model for predicting late winter $h_{SWE}$ comes from an overall comparison made with in situ measurements of glacier winter mass balance (= late winter snowpack depth) at reference stakes on Svalbard glaciers. Results of these comparisons are reported in van Pelt et al. (2012) as RMSE values. Note that these RMSE values, which range between 0.12 and 0.33 m w.e., may overestimate the true magnitude of the model errors, because they are based on comparisons of the 1 km$^2$ mean $h_{SWE}$ values produced by the model with individual mass balance stake data, some of which are known to record more snow accumulation that the local km$^2$-scale mean. This information has been added to section 2.4.2. in the revised manuscript. We used the results from van Pelt et al. to compute $\pm$ 1 σ errors (assuming an overall normal distributon of errors), which are now shown on Fig. 6 in the revised paper. See also our reply to the reviewer's comment about this section, further below.

*# Page 7: $C_{snow}^{EC}$ air is used for some data interpretation. Here, no critical evaluation of this data is given. Single parameter MAC is used to derive this quantity. However, no basis is given to show that this methodology is supported. In addition, different instruments are used to derive this quantity. Associated uncertainties should be given.*

In the manuscript, the BC-equivalent mixing ratios at Zeppelin observatory ($C_{air}^{eBC}$) were used for two different purposes. On Fig. 5, the data were simply used to illustrate when the surface snow samples were collected relative to the seasonal cycle of eBC aerosols (panel e). Here, all that matters is the overall temporal pattern. In the revised manuscript, we have chosen to replace the eBC data on Fig. 5 with the aerosol absorption coefficient (what was actually measured). In section 4.2 of the original manuscript, we had also used to $C_{air}^{eBC}$ in an attempt to estimate a plausible range of scavenging ratios that could produce the observed EC concentrations in snow ($C_{snow}^{EC}$). We have, however, to remove this part of our analysis entirely in the revised paper.

*# Page 8. Table 2. It will be useful to include average altitudes for these sites to show altitude dependence of the quantities given in this table. Table 2 and Figure 5 will be most important in this paper. It will be useful if these locations are clearly shown in Figure 1, for example. There are a number of snow sampling locations given in Figure 1. It is difficult to relate them with Table 2 and Figure 5 for readers unfamiliar with the geography of Svalbard. You may consider using abbreviations, such as used in van Pelt et al. (2019).*

Table 2 (now Table 3) is just a summary of descriptive statistics for the main groups of observations presented in the paper. In the case of glaciers, our data come from a wide range of elevations, so including these in the table of statistics would not really be helpful because the altitude ranges for different glaciers largely overlap. Our Tables S1 and S2 give the altitudes of every site from which samples were obtained in this study. However we can not easily condense *all* the information from these two tables in a single one. As a compromise, we created a new table (Table 1 in the revised manuscript) that only presents the *geographic location information* (lat., long., and altitude) of the sampling sites. Other information, such as sampling dates, remain in Tables S1 and S2. We also did, in fact, used site abbreviations throughout the text (e.g, HDF, LF), but we used a different notation than that in van Pelt et al. (2019) because we had more sites to consider than they did. In retrospect, it would have been better to "streamline" these abbreviations, but as they are also now used in a companion paper to our own (also submitted to ACP), we prefer to leave them as they are.

*$L^{EC}_{snow}$ snow is almost proportional to $h_{SWE}$ up to $h_{SWE}$ = 800 mm. Apparently, the relation becomes non-linear due to the two data points at $h_{SWE}$ > 1200 mm. There is no physical basis to fit the data by an exponential function. In the linear region of $h_{SWE}$ < 800 mm, the slope gives an average CEC snow for this data set. I consider that $C^{EC}_{snow}$ for $h_{SWE}$ < 800 mm is constant as a first approximation, especially considering that the uncertainty is not well defined. It is likely this slope agrees with the average $C^{EC}_{snow}$ of individual sample values and this should be mentioned.*

The reviewer made an astute observation. When we first investigated the relationship between $h_{SWE}$ and EC (or WIOC) loadings in the snowpits, we were opened to the possibility that the relationship might not be linear, because neither EC nor WIOC concentrations in snow are constant, so their cumulative sum (adding all layers) might not scale up as a simple linear function of $h_{SWE}$, not if there was, for example, any sort of gradual change in EC or WIOC content of snowfall over the course of the snow accumulation period. When we re-examined the data which were used on the regression models shown on Fig. 6, we came to question the two largest values, i.e the two "high" points on Fig. 6. To explain: in our study, the samples that showed the strongest indications of dust (coloration or visible particles on uncombusted filters) came from the bottom layers of the seasonal snowpack on glaciers of southern Spistbergen (Hansbreen, Werenskioldbreen). Furthermore, the darkest filters came from snowpits that were excavated in the accumulation area of the glaciers (sites HB3 and WSB3), where the seasonal snowpack is underlain by firn (and not ice). This raises two possibilities. One is that some of these deeper snow layers are in fact firn formed in the previous summer on which dust and BC particles concentrated by ablation, and which were incorrectly identifed as being autumn snow layers in the stratigraphy (This had in fact been discussed while writing the manuscript). Another possibility is that dust and BC particles became aggregated and migrated towards, the base of the snowpack during thaw events after the onset of autumn snow accumulation. In the former case, the EC measured in these deep layers should be excluded in the winter 2015-16 winter burden, and in the latter case they should included. The former possibility, i.e., basal layers being firn rather than autumn snow, seemed the more likely one, as neither BC nor dust particles are readily mobilized into deeper snow during thaw. Furthermore if we assumed that the bottom layers in the snowpits at HB3 and WSB3 were in fact part of the previous summer firn, it brought the observed physical stratigraphy of the snowpits in better agreement with what was expected based on the simulated sequence of snow accumulation and thaw events (the output of the snowpack model). We therefore excluded the two deepest snowpit layer samples with the higher EC concentrations and recalculated $L^{EC}_{snow}$ and $L^{WIOC}_{snow}$ for the snowpits at sites HB3 and WSB3. In doing so, we weighted the regressions inversely with the estimated uncertainties of $L^{EC}_{snow}$ and $L^{WIOC}_{snow}$, as we had done before. The results (Fig. 6 in the revised manuscript) show that, indeed, the relationships with $h_{SWE}$ are closer to linearity (but not constant). This is certainly true for $L^{EC}_{snow}$, but less clear for $L^{WIOC}_{snow}$, as these data show greater scatter. Our estimates of the overall deposition of EC and OC across Svalbard were updated accordingly in the text and in Fig. 7 of the revised manuscript. On account of excluding observations from the deepest layers at sites HB3 and WSB3, we also updated other plots (e.g., Fig. 3 and 4) and some of the descriptive statistics (Table 3). However, none of these changes have any incidence on the other main conclusions of the paper.

*# Section 4.1, page 9-10. Spatial and temporal variabilities of EC and OC are discussed here. It is difficult to follow them in a short time due to many numbers presented here. This part should be re-structured, may be by using a figure or table. Using abbreviations may also help. Details should be moved to Supplement.*

This section has been restructured and considerably simplified in the revised version of the paper. Unessential details were removed, and clear references to the various sampling sites (using both names and letter codes) have been added in the text.

*# Section 4.1, page 12, L354-365 and page 15, L 460: Two data points at hSWE > 1200 mm are interpreted as the result of inflow of polluted air from south associated with large cyclonic storms. Only LEC snow is discussed here, but it is the linear slopes of these points from the origin (CEC snow), that are most relevant here. It will be useful to check if CeBC air at Zeppelin showed large increases during these events. It should be enhanced by large factors if air transported from south was polluted. It is stated that storms and landfall caused large LEC snow. It may explain large hSWE, but it will not explain large CEC snow. The effects of storms on hSWE and CEC snow should not be mixed*

This is valid point. We have considerably simplified this part of the discussion and removed this more speculative aspect of it.

*# Section 4.2 and other parts. Variation of* snow $C_{snow}^{EC}$ *with altitude is an important point, again depending on the reliability of the data. Considering its importance, more detailed discussion on possible causes of this feature should be given.*

This comment is unclear: What "feature" is the reviewer referring to ? As discussed at length above, we are confident that the lack of obvious differences in the median $C_{snow}^{EC}$ with elevation is not due to the lack of reliability in the data. The fact that we did not find any significant differences does not exclude the possibility that there may, in fact, be some gradient in EC *deposition* in snowfall with altitude in Svalbard. It simply means that the spatial and temporal variability of $C_{snow}^{EC}$ *in snow on the ground* is such that we can not presently detect such patterns (although we do find observe gradient in net EC accumulation in the snowpack). However, it should be noted that almost all land areas of Svalbard (with the possible exceptions of a few high points on central Spitsbergen) are probably within the planetary boundary layer (PBL) during winter (see Fig. 2), so it could simply well be that mixing of air within the PBL obscures any elevation-dependence for EC and WIOC in snow.

*# Section 4.2 and other parts. The effect of dry deposition is considered to be small from the analysis Figure 6. I agree with this analysis. However, the effect of dry deposition may be substantial for other snow samples. More considerations should be made on this point.*

This suggestion is vague. Obviously, our analysis is based on a limited set of observations, our own, and the interpretation that we present are based on these data alone. We are aware that other studies offer different conclusions on the possible importance of dry deposition (for e.g., Jacobi et al., 2019). We make no claim that our study gives a definitive answer on this issue. But we see no point either in engaging is a speculative discussion about what might happen under other circumstances for which we have no data.

*# Section 4.2, page 13, L 398-427. The analysis made here assumes that $C_{air}^{eBC}$ at the surface is directly linked with $C_{snow}^{EC}$ snow. No detailed analysis is made to support this assumption and the discussion based on this assumption is very weak. It does not add solid materials to this paper. Therefore, it should be deleted. A minor comment. W is not non-dimensional. A proper unit should be given to it.*

We have opted to remove this part of the discussion in the revised paper.

*# Section 4.3. In this section, CEC snow data from different groups using different TOT methods are collected. There are two problems here. In this part, no detailed analysis and discussion are made on the uncertainties of EC data by different groups. It is not scientifically sound to simply collect the data without any critical evaluations. A second problem is that this section has little relevance with the major points of this paper and deviates from the mainstream of this paper. The circum-Arctic data are best analyzed and discussed in detail as a separate paper. Therefore, this section should be deleted.*

The reviewer questions whether the presentation of the geographic comparison of results (our Fig. 9) fits within the paper or not. We consider that it does. To place our findings in a broader geographical context is entirely legitimate and justifiable, and is very commonly done in these types of surveys. The graphical manner in which we compare data from multiple sites (using box plots of the probability distributions) is also a more *prudent* way to do this than by simply comparing mean values, as it too often done, and which can be very misleading indeed. Therefore, we have kept this section in the revised manuscript. However, we have shortened and de-emphasized it, and have also considerably simplified (as well as corrected errors in) our revised figure (Fig. 10).

The other point raised by the reviewer is similar to the one discussed earlier, i.e., whether the general pattern of geographic variations seen on Fig. 10 is real, considering possible methodological differences in the datasets. All the $C_{snow}^{EC}$ data shown in the original version of the paper were produced by either one of two thermo-optical protocols, either NIOSH 5040 (introduced in 2003) or EUSAAR-2 (introduced in 2010), with the exception of the data from Alaska, which were produced with the DRI protocol. In our revised version of the figure, we have now removed the Alaska data, because there are no intercomparisons of this protocol with the NIOSH 5040 or EUSAAR-2, whereas differences between the latter two protocols were evaluated in several studies (e.g., Cheng et al., 2014). On the basis of these studies, the data in Fig. 10 that had been obtained by the NIOSH 5040 protocol were corrected by a factor of two, as was discussed in the manuscript.

However, variations in $C_{snow}^{EC}$ may still arise because of different instrument settings used between studies, for example the choice of the reflectance (TOR) or transmittance (TOT) as the optical strategy to correct for charring effects. A useful indication of the possible spread of results comes from the intercomparison study by Panteliadis et al. (2015), in which a common set of aerosol filters were analyzed by 17 institutions using NIOSH or EUSAAR-2 protocols on Sunset ECOC analyzers, but with varying instrumental setttings and with both TOT and TOR corrections. The filters analyzed in this study had EC mass densities of ~1-15 mg cm$^{-2}$. These are quite typical for the snow filters analyzed in our study, and would translate to $C_{snow}^{EC}$ of ~0.5-10 ng g$^{-1}$ with the snow sample volumes used. Overall, Panteliadis *et al.* (2015) found the reproducibility and repeatability of EC analyses (across all methological variations) to be between 15-20 % for EUSAAR-2 and between 20-26 % for NIOSH. Very simular results were obtained by Bautista et al. (2015), who compared results of EC analyses in filtered aerosols by the NIOSH, IMPROVE_A and EUSARR2 thermo-optical protocols, using both TOT and TOR charring corrections. To our knowldege, no equivalent inter-comparison studies have been published for EC measurements in snow, but it seems unlikely that results would be vastly different, as the thermal desorption principle applies in the same way to particulates filtered from air or from meltwater (both containing mixtures of BC, mineral and organic phases).

On our new figure (Fig. 10), we show, in grey shading, the estimated spread (interquartile range) of $C_{snow}^{EC}$ values that might be expected to occur due to analytical method inconsistencies for median $C_{snow}^{EC}$ of ~3, 10 and 12 ng g$^{-1}$, based on the variability of results obtained by Panteliadis *et al.* (2015) and Bautista *et al.* (2015) for aerosols. It can be seen that most of the differences in the median $C_{snow}^{EC}$ seen across regions are larger than the plausible spread of medians that might arise due to methodological inconsitencies in EUSAAR_2 and NIOSH. The large-scale variations in $C_{snow}^{EC}$ between different sectors of the circumarctic shown on Fig. 9 are therefore unlikely to be due to uncertainties of the TOT method. As before, what mostly limit our ability to confidently resolve geographic variations in $C_{snow}^{EC}$ is the inherently large variability in these data, rather than the methodological uncertainties.

*Numerical values of parameters such as $C_{snow}^{EC}$, $C_{snow}^{OC}$, $L_{snow}^{EC}$, and $L_{snow}^{OC}$ snow are given in 3-5 digits at many places, including abstract. Considering that the uncertainties cannot be given and are potentially large, these numbers should be given in 2 or 3 digits.*

This has been corrected throughout the paper and in all tables and figures.

*# Page 2, L45: BrC is referred to here. But no discussion is made in the discussion of OC in this paper.*

This was removed altogether in the revised paper.

*# Page 14, Line 413: "f" is not clearly defined. Is it zero or one ?*

This was removed altogether in the revised paper.

*# Page 16, Line 484: UL may be UW.*

Corrected.

**References cited**

Bautista et al. (2015) doi:10.5094/APR.2015.037.
Cheng et al. (2014) doi:10.1016/j.scitotenv.2013.08.084
Jacobi et al. (2019) doi:10.5194/acp-19-10361-2019.
Panteliadis et al. (2015) doi:10.5194/amt-8-779-2015
van Pelt et al. (2012) doi:10.5194/tc-6-641-2012.

**acp-2020-491**
**Spatiotemporal variability of elemental and organic carbon in Svalbard snow during 2007-2018**

**Author's response to specific comments**

**Reviewer 2**

*L42: While it's importance is realized for climate forcing, it is understood that BrC is the kind of organic carbon, including water-soluble organic carbon (i.e., WSOC), mostly polar organic compounds (e.g., oxygenated OC or organic nitrogen compounds or organosulfur compounds). However, the OC in this manuscript does not include the fraction of WSOC, which was lost by ~ 80% through the filtration process. Please re-write this part accordingly and relate BrC to the water insoluble OC as WIOC (e.g., solvent soluble OC are also part of BrC).*

In the revised paper, we have adopted the WIOC terminology as suggested, and we no longer refer to BrC. We have limited greatly our discussion of possible nature and sources of WIOC, as this is not the main focus of the paper.

*L60-65: There are some confusion here since this study followed the filtration method by Forsstrom et al (2009, 2013) and utilized the EC/OC analysis by EUSAAR_2 but providing the citation by Chow et al, 2004. Please clarify.*

The correct reference is now provided.

*L66: "OC" should be defined and clarified here*

*Corrected in the revised text.*

*L70-71: I would prefer "scavenging ratio" to "washout ratio".*

We have opted to remove this part of the discussion altogether.

*L75: it would be better replacing the section title with "2.1.1 April 2016 glacier survey"*

In the restructured new version of the paper, the section on the Spring 2016 glacier survey is cleary identified and separated from other data.

*L84: Fig. 2 is hard to follow; particularly for the caption, (the color in lower panel does not match these inupper panel). Please modify the figure to make it easier to understand or take it out (which not belongs to the key figures).*

This figure has been simplified (only one hypsometric curve is now shown) and we have added on it labels showing the distribution of the various snow sampling sites with respect to elevation.

*L87: please replace "stable oxygen isotope ratio ($\delta^{18}O$)" with "stable oxygen isotope ratio ($\delta^{18}O$) in water"*

Corrected.

*L104: Please spell out 'ERA' for the first time.*

Clarified in the revised text.

*L111: Suggest modifying the section title to "Surface snow monitoring (2007-2018), Brøgger Peninsula"*

Accepted as suggested.

*L125: These additional samples in Table S2 are confusing as the sampling sites and the date are partially overlapped with these samples in Table S1. Are there any special purpose for those samples in Table S2? If not, it make more sense to include them in section 2.1.1 and change the current title to "2.1.1 2016-2017 glacier survey". Suggest to make 2 sub-sections: one is about the major survey in April 2016 and the other is about the additional irregularly sampling by NPI staff (2016-2017).*

We have clarified this in the revised manuscript by describing the Spring 2016 and 2017 glacier survey data separately.

*L130-140: As for EC and OC analysis of the snow samples, there are several steps involved, including snow processing (melting & filtration of particles, i.e., EC & WIOC on to filters) and EC/OC analysis. Large amount uncertainties would associated with these steps and the uncertainties of the snow processing are unknown largely. Author stated following the procedure by Forsstrom et al., 2009. However, the procedure by Forsstrom et al., 2009 was only for EC and the procedure regarding how to deal with the WIOC was not described, which is very important to this study. Author should provide more details of description for the snow processing steps, e.g., how many minutes were used for melting snow? and what kind of device was used for filtration of snow ? and how many minutes were used for the filtration? It is suggested to use some proxy of reference EC and WIOC (mentioned above) and ionized water to get the recovery rates for the filtration process.*

We refer the reviewer to the discussion above. We added much more complete descriptions and details about the methods and its potential uncertainties in sections 2 and 4 of the revised manuscript.

*L 136: The definition of OC mass measured (i.e., WIOC) should be clarified here also.*

This has been clarified as recommended.

*L141-153: It is suggested that more detailed error analysis is added here, as mentioned in general comments.*

See discussion above, the descrption of sources errors in section 2.2.2. (which were formerly in the Supplement), and the discussion on the contributions of these errors to the observed variations in snow, in section 4.2.1.

*L67: Please modify the section title to "2.3 $\delta^{18}O$ analysis in snow water"*

*Modified as suggested.*

*L168: Please replace the expression of "The stable isotope ratio of oxygen (16O:18O)" with "... (18O:16O)".*

Corrected.

*L167-172: A sentence should be included here regarding why the authors should include $\delta^{18}O$ data in the study (Fig. 8). Otherwise, the $\delta^{18}O$ should be moved to "supplementary materials", together with all support data. Thus, the main theme could be better presented without detraction from those supporting data.*

This was in fact clearly specified in the original manuscript, section 2.3: The stable isotope ratio of oxygen ($^{18}O$:$^{16}O$) in snowpit samples collected in April 2016 was used to detect evidence of warming events associated with large autumn or winter snowfalls, that could help to interpret the $L_{snow}^{EC}$ and $L_{snow}^{WIOC}$ data.

*L173-L216: It is suggested to move the content in section 2.4 to "supplementary materials" to make the theme stick out.*

We have kept section 2.4 as it is, because the snowpack modeling was an important element of the study methods.

*L219: Please use plain language to explain "skewness" as a statistic concept. What does it mean applying to your data?*

To clarify, we have reformulated this passage as follows: "The probability distributions of $C_{snow}^{EC}$ and $C_{snow}^{WIOC}$ are positively-skewed (right-tailed), therefore we use medians ($\tilde{C}_{snow}^{EC}$, $\tilde{C}_{snow}^{WIOC}$) as measures of their central tendency (...)".

*L218-L230: This paragraph is full of numbers and hard to follow what the author would like to express. It is suggested either totally deleting this paragraph or focusing on the description of data structure including skewness and LOD and how would the data structure affect the interpretation of this dataset and the comparison it with other studies.*

We consider that providing clarifications on how measures of central tendency were calculated are important when dealing with skewed data with values < LOD, so we have left these details in the text. However, the paragraph as a whole has been shortened and simplified, with fewer numbers.

*L231: L75: Please modify the current title to "3.1 April 2016 glacier survey", being consistent with 2.1.1.*

Changed as suggested.

*L261-270: It is suggested to include all the annual median concentration data (i.e., $C_{snow}$ on EC and OC) from both Ny-Alesund and Austre Broggerbreen shown on Fig. 5c in a table, in order to better understand the paragraph.*

These are now displayed on the revised Fig. 5, for greater clarity.

*L272: Ensure that the section title be consistent with 2.1.1 and 3.1*

Changed as suggested.

*L273: the general statement for "no discernible zonal or latitudinal gradient of $C_{snow}$ on EC or OC across Svalbard" is not convincing, as the uncertainties are not well investigated.*

See discussion above, in particular paragraphs 5 to 16.

*L278-279: highest among what ? This sentence is not clearly expressed. In fact, there is not much difference shown on Fig. 3 and 4.*

This passage has been deleted altogether in the revised paper.

*L282-283: This statement seems not supporting by the data shown in Figs, S4-S5, and Figs S8-S9. As suggested, it is better to table all the data in the Supplementary Material to see clearly.*

See response to previous comment. Figs S4-S5 and S8-S9 were removed from the Supplement in the revised version, as it was felt these were not essential and tended to create unnecessary confusion.

*L299-L302: Please check the numbers mentioned in L299-L302, which are not consistent with those in Fig 6a and 6b.*

The modeled relationships of $L_{snow}^{EC}$ and $L_{snow}^{WIOC}$ with $h_{SWE}$ were modified following a comment from reviewer 1 (see above) and consequently both the text and Fig. 6 have also been modified.

*L272-365: The Section 4.1 is too long to follow, covering the discussion on different topics, including*
*- surface $C_{snow}$ of EC and OC*
*- Lsnow on EC & OC and the relationship between $L_{snow}$ and $h_{SWE}$*
*- Snowpack modeling and simulated $h_{SWE}$, and $L_{snow}$ (via extrapolating the relations shown in Fig. 6)*
*- Simulated total EC and OC accumulated mass during the winter of 2015-2016 (Sept. 2015-April*
  *2016) across Svalbard and derived monthly and daily deposition rate in the area.*
*- Simulated snowpack profiles and related them to the cumulated mass of EC and OC (i.e., $L_{snow}$ of*
*EC and OC) shown in Fig. 8 and Fig. 9 and discussion snow dynamics impacts.*
*I would suggest to re-arrange those contents in one section 4.1 into several sub-sections as 4.1.1, 4.1.2, 4.2.3 etc.*
*accordingly based on topics mentioned above. Author may consider moving all snowpack modeling & snow dynamics*
*related contents to the Supplementary Materials and briefly include the most relevant results here accordingly to make the*
*main theme clear.*

Section 4 has been restructured more clearly and considerably shortened. The part that discuss the relationships of $L_{snow}^{EC}$ and $L_{snow}^{WIOC}$ with $h_{SWE}$ and the extrapolations from these models, and the one that discusses the relative timing of EC and WIOC accumulation in the winter 2015-16 snow have now been placed in dedicated sections.

*L366 -427: Similar to section 4.1, suggest to re-arrange this section 4.2 into two sub-sections as:*
*- L367-397: 4.2.1 "Temporal variation of $C_{snow}$ on EC and OC"*
*- L398-427: 4.2.2 "Scavenging rate of EC, Brøgger Peninsula"*

This section has also been considerably shortened and simplified, and the part in which EC scavenging ratios were estimated from aerosol data has been entirely removed.

*L395-397: It is suggested to begin with the sentence via using "To understand the possible role snowfall anomalies over center Brøgger Peninsula..." instead of "To control for the possible role of snowfall rate…"*

Modified as suggested

*L429: Suggested to modify the section title as "Comparison of $C_{snow}$ EC in Pan-arctic perspective".*

Modified as suggested

*P34, Fig. 9 All the sites labelled in the plots here should be included in a map identified each of them.*
*ALB3 could not found in Fig. 1.*

We have actually chosen to simplifiy this figure considerably, so the Svalbard $C_{snow}^{EC}$ data are now shown in fewer categories, thus avoiding unnecessary confusion, as we only intended to highlight the broad geographic patterns.

In Supplementary Section:

*Under "EC and OC analyses: Additional information": Uncertainties estimation and analysis should be paid more attention and include more discussion as mentioned in the General Comments.*

See discussion above. The details on methods that were previously in the Supplement were moved into the main text, and the discussion of these errors expanded.

In addition to the uncertainties of $\sigma_{sh}$, $\sigma_u$ and $\sigma_f$, for EC, there are $\sigma_{ue}$ (Under Estimate) due to dust (e.g., $Fe_2O_3$) and $\sigma_{oe}$ (Over Estimate) due to carbonate; whereas for OC, there are $\sigma_{lwsoc}$ , i.e., loss of WSOC ($\sim > 80\%$ of total OC, Hagler et al., 2007) during filtration processing, and $\sigma_{oe}$ due to dust and carbonate. The overall uncertainties should be derived via error propagation.

See discussion above, and particularly paragraphs 16 to 24. Parts of this discussion were integrated in the revised paper either in section 2 (methods) or section 4 (discussion). We chose to de-emphasize the OC results in a paper, and we now designate these data as WIOC, as was suggested. The errors in the $C_{snow}^{EC}$ that may arise due to the presence of dust are presently very difficult to quantify. Our review of the published literature on this topic was not helpful, as different studies give different conclusions, for e.g., whether does dust leads to under- or over-estimation of EC, and if so, by how much. The most directly relevant results for the present study are those of Svensson et al. (2018), because the methods they used were in almost every detail the same as ours. Their study suggests that the effect of dust may not, in fact, be a major concern at the moderate concentrations found in Arctic snow, as least for thermo-optical EC analysis (optical methods are a different matter altogether). As we discuss in section 4.3. of the revised paper, there are simply insuffcent consistent data to quantify dust-related errors in our data. If we attempted it, it would be largely guesswork, so we abstained. We have instead endeavoured to quantify the possible effect of those method-related uncertainties that are reasonably known, and discuss the possible effects of dust in a qualitative, rather than quantitative, manner.

*Fig. S2: Please show the relative location of Fig. S2 to Fig. 1.*

This is shown by a framed area on Fig. 1 near Ny-Ålesund, and is specified in the revised caption for Fig. 1.

*Fig. S4-S5: Since the log scale is used, it looks no much variation observed. In fact, large variation may exists. It is suggested to list all the data in a table.*

These figures have actually been removed altogether from the revised Supplement.

**acp-2020-491**
**Spatiotemporal variability of elemental and organic carbon in Svalbard snow during 2007-2018**

**Summary of main changes made to the manuscript**

- The title was changed to: "Elemental and water-insoluble organic carbon in Svalbard snow during 2007-2018: A synthesis of observations during 2007–2018"
- As suggested, we adopted "water-insoluble organic carbon" (WIOC) instead of OC throughout the paper.
- To clarify the identification of the various sampling sites, letter codes were added to the general location map (Fig. 1), and a new table (Table 1 in revised ms.) was added that lists all sites. Other clarifications were made in the text.
- Additional details were added on the sample collection and preparation protocols (in section 2) and a new figure (Fig. S1 in revised ms.) was added in the Supplement, showing the filtration apparatus.
- The discussion of methodological uncertainties was greatly expanded. The part that was previously in the supplement was moved into the paper. Uncertainties related to sample collection, preparation and filtration steps are, as before, discussed in section 2 (Methods), while uncertainties that may arise due to the presence of dust in snow are discussed in section 5 (Discussion). This split is to avoid making the text too "front-heavy".
- We estimated the probable contribution of methodological uncertainties to the overall variation of results, and how this may affect the main conclusions regarding spatial and temporal variations. This is described at length in the discussion below, and was also integrated, in an abbreviated format, in section 5 of the paper (Discussion).
Two figures pertaining to this were added in the Supplement (Fig. S5 and S6).
- Throughout the paper, we have de-emphasized the discussion of WIOC results because the uncertainties on those measurements are more poorly constrained than those of EC. However, we keep the WIOC data as part of the paper because we consider it to be valuable, albeit ancillary, information.
- Errors were found in the descriptive statistics (Table 2) due to some data points being mislabelled and double-counted in two categories, and these errors have been corrected. The corrections have no effect on the figures or on the main findings, however.
- The estimates of EC and WIOC loadings across Svalbard based on regression (in section 4.1) were revised based on suggestions from the reviewers.
- The part of section 4.2 from the original manuscript in which we estimated BC scavenging ratios from aerosol and snow data was removed entirely.
- The part of section 4 that compares data from Svalbard to data from other circum-Arctic sites remains, but it has been simplified, shortened, and de-emphasized.
- To increase clarity overall, several figures were simplified (removing excess information).
- Some figures in the Supplement, judged unessential, were removed altogether.

---

## Author Response (AR2)

"Elemental and and water-insoluble organic carbon in Svalbard snow: A synthesis of observations during 2007-2018." by Zdanowicz *et al.*

**Authors' reply to comments by Reviewer 1**

*Major comments*

*[1] The uncertainties of the present method are described separately in sections 2.2.2. and 4.3. The uncertainties should be given before the observational results are presented. Otherwise readers need to re-consider or re-interpret results that are shown before section 4.4. These sections should be combined together as much as the logical structure of the text allows.*

This is a matter of author or editorial choice. For example in their 2010 paper published in ACP, Doherty et al. placed their ~5-page long analysis of measurement uncertainties in section 6, after their presentation of results. However our own discussion of the uncertainties due to the presence of dust, is shorter, so we have now moved it into the Methods section in the revised manuscript, as suggested by the reviewer.

*[2] Figure 4: Probably this figure is most important in this paper. It will be useful to show the vertical profiles with linear-scales, at least for the median values, in addition to the log-scale plots.*

Added to Fig 4, as suggested.

*[3] Page 14, L409-412: It is stated "of particular interest". However, explanations of this "interesting" feature are very poor.*
*a) There are no explanations why this feature is particularly interesting.*

The features in the $C_{snow}^{EC}$ and $C_{snow}^{WIOC}$ data are interesting (or noteworthy, if one prefers) in the sense that they depart from invariance or from homogeneity, and they appear to be temporarily consistent over the years. We have made this more explicit in the opening of section 4.3 in the revised text. Concerning the *plausibility* of the proposed explanations for these feures, see below.

*b) L411, Is the "gradient in EC and WIOC" in horizontal or vertical directions?*

Both are of course possible, i.e. the apparent systematic difference in the median $C_{snow}^{EC}$ and $C_{snow}^{WIOC}$ between Ny Ålesund and the Austre Broggerbreen site (which are ~5.5 km apart) might be linked to the relative horizontal distance from Ny Ålesund and/or from the coast (i.e. transport distance from local EC or WIOC sources), or to the difference in elevation. The two go together, since air moving inland from the coast necessarily rises over the interior plateau. We have nadded some nuances to the text to this effect.

*c) There is no interpretation on how localized EC emissions impact EC in snowpack at Ny-Alesund. The effects of in-cloud scavenging, below cloud scavenging, and dry deposition should be discussed with some quantitative analysis.*

To perform the sort of quantitative analysis the reviewer suggests, one needs, ideally, simulteaneous measurements of BC (or eBC) in air and of BC (or EC) in falling snow, or at least in freshly deposited snow (e.g., Sinha et al., 2018). The snow samples collected near Ny Ålesund between 2007-2018 were obtained (as described in the methods) in an opportunistic way, whenever staff at the NPI station were available to do it. While on a few occasions the surface snow was sampled shortly after snowfall events, this was far from being always the case. Furthermore, it is difficult to ascertain what time

interval of snow accumulation is represented by each surface snow layer: one can not assume linear accumulation over time, since snowfall is irregular and there is surface wind drift. This makes a direct comparison of airborne and snow BC (EC) data highly uncertain (see also 4th paragraph, below).

In a relevant study, Jacobi et al. (2019) combined optical measurements of eBC from Zeppelin station with snowpit measurements of rBC at high depth resolution (~3 cm intervals) made at a comparable altitudes on two glaciers (Kongsvegen and Austre Lovénbreen). They used the CROCUS snow model to estimate, indirectly, the temporal sequence of snow accumulation at the snowpit site, and from these data, they computed the predicted monthly cumulated BC deposition in snow by both and wet dry deposition, which they then compared with the rBC burden in the snowpit. However we can not reproduce such an analysis since our own snowpit data have a much coarser depth resolution (imposed by the sample volume requirements for TOT analysis) which can not easily be related to specific snowfall events.

Another alternative is to use some indirect, model-based estimates of atmospheric BC column loadings in combination with data on the vertical cloud structure to compute BC deposition in snow. During the preparation of this manuscript, we considered, at one point, using modeled vertical BC aerosol column loadings over Ny Ålesund produced by the CAMS global reanalysis (e.g., Pakszys & Zielinski, 2017) to estimate both dry and wet deposition and compare these results with the EC concentrations in snow. However we found, upon verification, that there was a poor match between the CAMS predicted near-surface BC mixing ratios at Zeppelin and the measured eBC, so we concluded that the CAMS reanalysis for BC were not sufficiently reliable at this latitude to be used.

In the first version of this manuscript (which the reviewer may not have seen), we actually made an attempt to relate eBC concentrations in air measured at the Zeppelin station to EC concentrations in surface snow on Austre Brøggerbreen (~same altitude as Zeppelin). We could not do the same at Ny Ålesund since we do not have aerosol data from this lower-altitude site. For Austre Brøggerbreen, we assumed dry deposition in snow to be minor or negligible (which is what our various glacier snowpit data suggest) and tested a range of plausible eBC scavenging ratios, combined with meteorologically-driven model estimates of surface snow accumulation rates, to find out which values yielded eBC concentrations in snow that most closely agreed with our measured EC concentrations. However, reviewers of the manuscript criticized this exercise to be too speculative and uncertain, partly owing to the fact that eBC and EC are not the same fractions of BC. We therefore chose to abstain from this sort of quantitative analysis in the revised manuscript, and we stand by this decision.

Our data are a survey of the spatial variability of EC and WIOC in snow across Svalbard. We observe, in these data, some apparent features for which we offer some possible explanations. These explanations could be put to the test in future studies. The higher median EC concentrations in snow at Ny Ålesund compared to Austre Brøggerbreen suggest an influence of BC emissions from the town on local snow (although it does not mean that all EC in snow there is from local sources). This certainly seems plausible. To rigorously test whether this is in fact the case, one would need to collect, simultaneously, both BC aerosols and falling snow (or freshly fallen snow) over at least one winter at both sites. In the concluding statement to the manuscript, we suggest this as a future task for experimenters and modellers, but we refrain from attempting such a quantative analysis ourselves, since we simply do not have adequate supporting data to do so.

***d)*** *Higher EC at Ny-Alesund is interpreted as due to localized EC emissions and lower EC at ABG is due to the gradient in EC. Then why do EC values vary synchronously? If EC at ABG is relatively free from local emissions, the EC values at these sites should vary quite differently from those at Ny-Alesund.*

This implies a false dichotomy, i.e. that EC deposited in snow at Austre Brøggerbreen is free of *any* influence from Ny Ålesund BC emissions, and, conversely, that EC in snow in Ny Ålesund is *only* from local BC emissions. We have made no such claim. Rather, we proposed that the higher median EC concentration in Ny Ålesund could reflect the influence of local BC emissions at this site, and the lower median EC concentration at Austre Brøggerbreen suggest that this influence is weaker up there due to greater inland distance and/or higher elevation. One possible explanation for this difference is that BC emitted in Ny Ålesund may tend to be trapped by low-level wintertime thermal or humidity inversions below the elevation of Zeppelin observatory. This is certainly plausible, and was in fact one of the reasons the observatory was established on Zeppelin Mountain. The effect of winter inversions might also apply to WIOC emitted from either combustion sources (in Ny Ålesund) or from marine sources (in nearby coastal waters). None of this excludes the possibility that EC and WIOC from other, distant sources (continental or marine) reach these sites, as must certainly be the case, based on previous studies of aerosol provenance (e.g., Eleftheriadis et al., 2009). If the efficiency of long-range transport (or deposition) of these aerosols varies between different seasons, then it is entirely plausible that this would affect EC and WIOC deposition in snow in Ny Ålesund and on Austre Brøggerbreen simultaneously, as the two sites are relatively close to one another. But even in this was the case, there could still exist a gradient in EC and WIOC concentrations in snow between the two sites, owing to the added contribution of local aerosol emissions from Ny Ålesund and nearby waters.

The reviewer's comment, however, did highlight flaw in part of our discussion. We had speculated that changes in the strength of winter inversions might account for the apparent synchronicity of the interannual variations in $C_{snow}^{EC}$ and $C_{snow}^{WIOC}$ in Ny Ålesund and on Austre Brøggerbreen, the argument being that changes in the mean thickness of the winter boundary layer might allow, in some years, more of the locally-emitted EC and WIOC aerosols to reach higher up and be deposited in snow on Austre Brøggerbreen. If this were the case, one might expect that the difference in $C_{snow}^{EC}$ and $C_{snow}^{WIOC}$ between Ny Ålesund and on Austre Brøggerbreen would decrease at these times. To find out if this was the case, we plotted, on our revised Fig. 5(d), the ratios of $C_{snow}^{EC}$ and $C_{snow}^{WIOC}$ between the two sites. Results show that in fact, these ratios increased in seasons when $C_{snow}^{EC}$ and $C_{snow}^{WIOC}$ were relatively higher in both Ny Ålesund and on Austre Brøggerbreen. Thus, changes in the mixed layer thickness are unlikely to explain the simultaneous interannual variations in $C_{snow}^{EC}$ and $C_{snow}^{WIOC}$ at the two sites. Instead, these variations, assuming they are real, seem more likely to reflect interannual variations in transport of removal of aerosols (irrespective of sources) that simultaneously affect both sites. Accordingly, we have modified parts of our discussion to stress this.

Note: While reviewing the paper to answer the reviewer's comments, an error was discovered. In section 3.2., L304, we had stated that *"...$\tilde{C}_{snow}^{WIOC}$ was 7 times higher, but as much as 30 times higher..."* in Ny Ålesund, when compared to ABG. These figures were actually incorrect: An offset of a year had accidentally been introduced in the calculations. Once corrrected, the magnitude of the differences in $\tilde{C}_{snow}^{WIOC}$ between Ny Ålesund and ABG turned out to be much closer to those seen in $\tilde{C}_{snow}^{EC}$, averaging ~2-3 in most years. The error did not affect the temporal pattern, nor our findings about $\tilde{C}_{snow}^{EC}$. We have corrected the relevant statements in section 3.2., in the discussion, and in the conclusions.

Note also that we have made some changes to Fig. 5: Panels (c) and (d) in the previous version have now been merged in panel (c), and a new panel has been added, which shows variations in the ratios of $\tilde{C}_{snow}^{EC}$ and of $\tilde{C}_{snow}^{WIOC}$ between Ny Ålesund and ABG. We also removed shadings from panels (c) and (e): These were included in the previous figure to indicate the possible range of variations in some of the results depending on whether data below detection limits were included or excluded. However

co-authors commented that the shadings compromised the readability of the figure, and so, on their advice, these were removed to simplify the plots and improve their overall clarity.

*e) Regarding (d), it may be useful to investigate the relation of EC in snow at ABG with BC in air at Zappelin for understanding the source of EC in snowpack at ABG.*

See response to (c) above.

*f) As one of possible sources of WIOC, natural source is discussed. Some more discussion on a relative contribution of natural additional source of WIOC is desirable.*

Presumably, this comment refers to page 14, L419-424, where we discuss possible marine sources of WIOC. We have stressed these particular sources because they might possibly account for the higher median $C_{snow}^{WIOC}$ observed near Ny Ålesund compared to Austre Brøggerbreen, higher up and further inland. We do not know of any other important *natural* source of WIOC emissions that would have such a localized impact in winter (tundra soil and vegetation emissions of volatile or semi-volatile OC are likely very low to nil in these months). There may, of course, be WIOC aerosols contributed from natural terrestrial emissions at lower latitudes (e.g., mainland Europe) and transported to Svalbard in winter, but it is not obvious why these should have systematically higher median concentrations in surface snow in Ny Ålesund compared to Austre Brøggerbreen, year after year. There may also be *non-natural* (i.e., anthropogenic) WIOC sources in Ny Ålesund, for e.g., as a component of vehicular exhaust emissions. We have added a few lines to section 4.3. to acknowledge this.

*g) In the response [15] of the authors, it is claimed that synchronous variations support the reliability of the method to some extent. This is not logically correct. First, the uncertainty of the measurements must be established. Then, it should be judged if the observed variations are larger or smaller than the uncertainties.*

The statement in our response (paragaraph [15]) which the referee quotes was about *probability*, not accuracy. We merely pointed out that if the observed temporal variations at the two sites were only due to random errors (caused by one or more factors), it is *unlikely* that these errors would combine to produce the same pattern of temporal variations at the two sites over several years. While the similarity of temporal patterns at both sites does not *prove* that these variations are real, it makes it *more likely* that they are. On Fig. 5(c), the estimated uncertainties (2σ) on the seasonal median $C_{snow}^{EC}$ and $C_{snow}^{WIOC}$ are indicated as error bars. These show that the amplitude of the synchronous interannual variations seen in Ny Ålesund and on Austre Brøggerbreen are in fact larger than the uncertainties for more than half of the years in which we have data from both sites. Some rewording in the revised section 4.3. makes the points above more explicit. We can add nothing further on the subject.

*Specific comments*

*Pages 12, L365: "Sihna" should be "Sinha"*

Corrected.

*Page 16, L500: "other analytical methods". The methods should be stated concretely. And their accuracies need to be compared with that of this work.*

We have now expanded this brief statement from L500 into a full paragraph where we provide the requested details.

*Figure 2: The colors for S and NE are similar. May be some symbols in the vertical profile can be changes (+ or open circles).*

On the revised figure, the color for the NE sector has been changed to a paler shade to increase the constrast with that for the S sector.

**No changes were made to the supplement.**

**References cited**

Doherty et al. (2010) doi:10.5194/acp-10-11647-2010.
Eleftheriadis et al. (2009) doi:10.1029/2008GL035741.
Jacobi et al. (2019) doi:10.5194/acp-19-10361-2019.
Pakszys & Zielinski (2017) doi: 10.1016/j.oceano.2017.05.002.
Sinha et al. (2018) doi:10.1002/2017JD028027.